# ON THE LEARNABILITY OF WATERMARKS FOR LANGUAGE MODELS

**Chenchen Gu, Xiang Lisa Li, Percy Liang, Tatsunori Hashimoto**
Stanford University
{cygu, xlisali, thashim}@stanford.edu, pliang@cs.stanford.edu

## ABSTRACT

Watermarking of language model outputs enables statistical detection of model-generated text, which can mitigate harms and misuses of language models. Existing watermarking strategies operate by altering the decoder of an existing language model. In this paper, we ask whether language models can directly *learn* to generate watermarked text, which would have significant implications for the real-world deployment of watermarks. First, learned watermarks could be used to build open models that naturally generate watermarked text, enabling watermarking for open models, where users can control the decoding procedure. Second, if watermarking is used to determine the provenance of generated text, an adversary can hurt the reputation of a victim model by spoofing its watermark and generating damaging watermarked text. To investigate the learnability of watermarks, we propose watermark distillation, which trains a student model to behave like a teacher model that uses decoding-based watermarking. We test our approach on three decoding-based watermarking strategies and various hyperparameter settings, finding that models can learn to generate watermarked text with high detectability. We also find limitations to learnability, including the loss of watermarking capabilities under fine-tuning on normal text and high sample complexity when learning low-distortion watermarks.[1]

## 1 INTRODUCTION

As language models (LMs) become more capable and widely used, watermarking LM outputs becomes increasingly important to mitigate potential harms and misuses of LMs. Watermarking enables statistical detection of LM-generated text, which enables enforcing policies on LM usage, e.g., removing LM-generated disinformation from social media platforms or detecting academic dishonesty. Another proposed use case of watermarking is identifying the provenance of text, i.e., tracing text to the specific LM that generated it (Abdelnabi & Fritz, 2021; Kuditipudi et al., 2023).

Recent works have shown that it is possible for an LM provider to inject specific, known watermark signals into text using specialized decoding algorithms (Kirchenbauer et al., 2023a; Aaronson, 2023; Kuditipudi et al., 2023), but little is known about whether these watermarks are learnable by a model. The learnability of watermarks has significant implications for the real-world deployment of watermarks, as it could enable downstream applications and adversarial spoofing attacks.

In this work, we study the learnability of watermarks by studying *weights-based watermarking*, which involves learning parameters for a language model that cause it to generate watermarked text under its natural sampling distribution, without using a special decoding-time watermarking algorithm. Our investigation is motivated by its relevant implications for two applications: (i) developing watermarking for open language models and (ii) spoofing watermarks.

First, existing watermarking methods depend upon using a specialized decoding algorithm, making them too inflexible for open LMs. For open LMs, where the weights are released, a user can use an ordinary decoding algorithm and generate non-watermarked text, whether intentionally or not. We find that weights-based watermarking works with standard decoding strategies, removing the

---

[1]See `https://github.com/chenchenygu/watermark-learnability` for code and models.

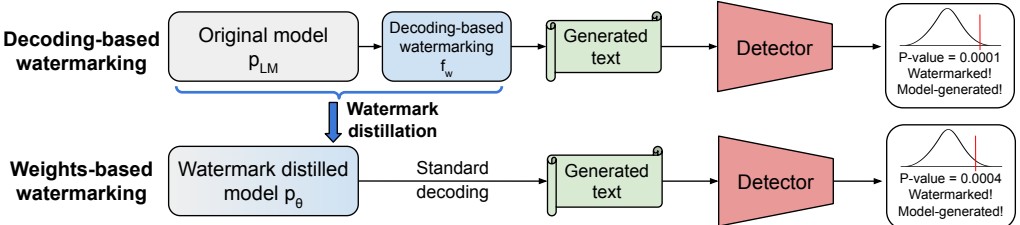

Figure 1: Decoding-based watermarking (top) versus weights-based watermarking (bottom). Decoding-based watermarking requires a specialized decoding algorithm $f_w$ to generate watermarked text, whereas weights-based watermarking can use standard decoding to generate watermarked text directly from the model, using just its weights. Watermark distillation enables weights-based watermarking by training a student model $p_\theta$ to behave like the teacher model $p_{LM}$ with decoding-based watermarking strategy $f_w$.

reliance on a specialized decoder. This makes it a promising first step towards developing watermarking for open LMs. However, we also find that weights-based watermarking capabilities can be removed by fine-tuning on normal text, indicating that improving robustness to fine-tuning is an important remaining challenge.

Second, in watermark spoofing attacks, an adversary outputs text that contains the watermark signal from a victim LM (Sadasivan et al., 2023). If watermarking is used to identify the provenance of text, then an attacker could attribute damaging text to the victim LM and hurt its reputation. We find that the learning of weights-based watermarking can enable spoofing attacks, and we demonstrate a proof-of-concept attack on an instruction-following chat model. The possibility of spoofing attacks suggests that watermarking should not be used to attribute provenance or blame to a specific LM. Instead, watermarking should only be used to statistically detect LM-generated text, which can be used for tasks such as finding infractions of policies on LM usage.

To enable weights-based watermarking, we propose logit-based and sampling-based watermark distillation, two simple methods for a student model to learn weights-based watermarking from a teacher model with decoding-based watermarking. Intuitively, in logit-based watermark distillation, the student model is trained to match the next token distribution outputted by the teacher model using decoding-based watermarking. In sampling-based watermark distillation, the teacher model with decoding-based watermarking is first used to generate watermarked samples. Then, the student model is fine-tuned on these watermarked samples.

We experiment with three decoding-based watermarking strategies: KGW (Kirchenbauer et al., 2023a; Zhao et al., 2023a), Aar (Aaronson, 2023), and KTH (Kuditipudi et al., 2023), and various values for their hyperparameters that control the level of distortion induced by watermarking. We find that watermarks and hyperparameter settings vary in their degree of learnability. In each watermarking strategy, higher-distortion hyperparameter settings are successfully learned by both forms of watermark distillation (median p-values less than 0.0001). Lower-distortion watermarks and hyperparameter settings are more challenging and less sample efficient to learn, but not unlearnable, as the p-values are still noticeably smaller than the non-watermarked baseline of 0.5.

## 2 BACKGROUND AND NOTATION: DECODING-BASED WATERMARKING

We study autoregressive language models $p_{LM} : \mathcal{V}^* \to \Delta(\mathcal{V})$ that map from a prefix string $x \in \mathcal{V}^*$ to a next token distribution over the vocabulary $\mathcal{V}$. Informally, a decoding-based watermarking strategy $f_w$ uses a watermark key $\xi$ to modify the model's original next token distribution $p_{LM}(\cdot \mid x)$ into a new distribution for generating watermarked text, which has a watermark signal embedded. The watermark detection algorithm $f_d$ looks for this signal using the same watermark key $\xi$.

Formally, we define a decoding-based watermarking strategy to be a function

$$f_w : \Delta(\mathcal{V}) \times \mathcal{V}^* \times \Xi \to \Delta(\mathcal{V}) \tag{1}$$

where $\Xi$ is the set of possible watermark keys. This function $f_w$ outputs a distribution $p_w(\cdot \mid x)$ from which to generate the next token in the watermarked text, given an original next token distribution $p_{LM}(\cdot \mid x)$, input text $x$, and watermark key $\xi \in \Xi$.

We define a watermark detection algorithm to be a function

$$f_d : \mathcal{V}^* \times \Xi \to [0,1]. \tag{2}$$

Given some text $x$ and watermark key $\xi$, $f_d$ outputs a p-value with respect to the null hypothesis that $x$ is independent of $f_w$ with key $\xi$. Informally, $f_d$ computes a test statistic that measures the strength of the watermark signal, then computes a p-value using the distribution of the test statistic under the null hypothesis. If the p-value is below a given significance level, the null hypothesis is rejected and the text is detected as watermarked. Slightly imprecisely, rejecting the null hypothesis means the text is detected as model-generated.[2]

In this paper, we consider three decoding-based watermarking strategies: Algorithm 2 in Kirchenbauer et al. (2023a), the Gumbel softmax scheme in Aaronson (2023), and the exponential minimum sampling scheme in Kuditipudi et al. (2023). Using the authors' names and initials, we refer to these as KGW, Aar, and KTH, respectively. We briefly describe these watermarking strategies below. See Appendix D for additional details and formal definitions.

**KGW: green list bias.** In the KGW watermarking strategy (Kirchenbauer et al., 2023a), when generating the next token, the vocabulary is pseudorandomly split into a "green list" and "red list" by hashing the previous token using the watermark key $\xi$. The green list contains watermark hyperparameter $\gamma \in (0,1)$ proportion of the vocabulary. Then, before the model's logits are converted to probabilities via the softmax function, hyperparameter $\delta > 0$ is added to the logits of the green list tokens. This procedure makes green list tokens more likely in watermarked text than in non-watermarked text. So, at detection time, if the proportion of green list tokens in a text is much greater than $\gamma$, then the p-value is small.

More generally, the previous $k$ tokens can be hashed, where $k$ is a hyperparameter. Values of $k > 1$ are investigated by Kirchenbauer et al. (2023b), finding that lower $k$ leads to more repetitive outputs. When $k = 0$, the green and red lists are fixed, regardless of the previous tokens. $k = 0$ was proposed by Zhao et al. (2023a) as Unigram-Watermark, a variant of KGW, but we will denote it as KGW $k = 0$ to simplify notation.

KGW distorts model outputs by upweighting green list tokens, increasing perplexity of generated texts computed by a larger model (Kirchenbauer et al., 2023a). Increasing the bias hyperparameter $\delta$ increases detectability, i.e., smaller p-values, but also increases distortion.

**Aar: boosting continuous hash scores.** The Aar watermarking strategy (Aaronson, 2023) hashes the previous $k$ tokens using key $\xi$ (where $k$ is a hyperparameter) to obtain a score $r_i$ for each token $i \in \mathcal{V}$, where each $r_i$ is uniformly distributed in $[0,1]$. Let $p_i$ be the original model probability for token $i$. Then, the next generated token is deterministically chosen to be the token $i$ which maximizes $r_i^{1/p_i}$, i.e., a token with both a high original probability $p_i$ and high hash score $r_i$. This procedure boosts the hash scores of tokens in watermarked text compared to non-watermarked text. So, at detection time, if the hash scores $r_i$ of the tokens in the observed sequence are high, then the p-value is low.

Since Aar deterministically selects the next token based on the previous $k$ tokens and the original model probabilities, Aar can lead to repetitive text, especially for small $k$ (Kuditipudi et al., 2023). Increasing $k$ decreases repetitiveness, as larger $k$-grams are less likely to be repeated, but the watermark also becomes less robust to edits, as each token edit affects the hash scores for $k + 1$ tokens.

**KTH: robust sequence alignment.** The KTH watermarking strategy (Kuditipudi et al., 2023) is similar to Aar, but instead of hashing previous tokens to obtain the scores $r_i$, the scores are obtained from the next element in the key sequence $\xi$. In KTH, $\xi = \xi^{(1)}, \ldots, \xi^{(m)}$ where each $\xi^{(j)} \in [0,1]^{|\mathcal{V}|}$

---

[2]This is slightly imprecise because model-generated text is not the only text that can be deliberately watermarked. For example, a human could write watermarked text by manually following the watermarking algorithm. However, for most practical use cases, such as detecting academic dishonesty, this minor imprecision is not an issue because either way, the user is doing something suspicious and unusual.

contains the scores, with entries uniformly distributed across $[0, 1]$. Then, to generate the $j$-th token in the sequence, KTH deterministically chooses the token $i$ that maximizes $\left(\xi_i^{(j)}\right)^{1/p_i}$. Note that $m$ should be larger than the maximum generation length. To allow different generations from the same prompt, before generating each sequence, $\xi$ can be shifted by some random $\tau$, i.e., $\xi' = \xi^{(1+\tau \bmod m)}, \ldots, \xi^{(m+\tau \bmod m)}$. To study the impact of these shifts on learnability, we introduce a hyperparameter $s \in [1, m]$ for how many shift values $\tau$ are possible.[3] Increasing $s$ expands the range of possible model generations.

At detection time, to be robust to text edits and shifts, the test statistic quantifies how well a text $x$ can be aligned with the key sequence $\xi$. More specifically, the test statistic computes a minimum Levenshtein distance using the alignment cost $d(x, \xi) = \sum_{t=1}^{\text{len}(x)} \log(1 - \xi_{x_t}^{(t)})$. A lower (more negative) test statistic indicates stronger watermark signal. To compute p-values, the observed test statistic is compared to a reference distribution of test statistics of non-watermarked texts. Letting $T$ be the number of samples in the reference distribution, the p-values computing using this method are lower bounded by $\frac{1}{T+1}$.

## 3 METHODS

**Problem statement.** Given a teacher model $p_{\text{LM}}$, decoding-based watermarking strategy $f_{\text{w}}$, and key $\xi$, the goal is to learn a student model $p_\theta$ whose sampling distribution naturally generates watermarked text. Specifically, letting $f_{\text{d}}$ be the detection algorithm corresponding to $f_{\text{w}}$, if $p_\theta$ generates text $y$ with small detection p-value $f_{\text{d}}(y, \xi)$ with probability similar to that of $p_{\text{LM}}$ with $f_{\text{w}}$, then $p_\theta$ has learned a *weights-based watermarking strategy*, since $p_\theta$ has learned to generate watermarked text using just its weights. Figure 1 illustrates decoding-based versus weights-based watermarking.

Next, we present two methods for learning a weights-based watermarking strategy: logit-based watermark distillation and sampling-based watermark distillation, which fall under the broader category of knowledge distillation (Hinton et al., 2015; Kim & Rush, 2016).

### 3.1 LOGIT-BASED WATERMARK DISTILLATION

In logit-based watermark distillation, we train the student model $p_\theta$ to behave as if it had decoding-based watermarking strategy $f_{\text{w}}$ applied. Specifically, given an input $x$, we want the student model's next token distribution $p_\theta(\cdot \mid x)$ to match $f_{\text{w}}(p_{\text{LM}}(\cdot \mid x), x, \xi)$, the next token distribution outputted by the teacher model $p_{\text{LM}}$ with decoding-based watermarking strategy $f_{\text{w}}$ and key $\xi$. So, given a dataset of texts $\mathcal{D}$, the training objective is to minimize the mean KL divergence between the teacher and student next token distributions on all prefixes in $\mathcal{D}$, given by

$$\mathcal{L}_{\text{logit}}(\theta) = \frac{1}{|\mathcal{D}|} \sum_{x \in \mathcal{D}} \sum_{t=1}^{\text{len}(x)} D_{\text{KL}}\left(f_{\text{w}}\left(p_{\text{LM}}\left(\cdot \mid x_{<t}\right), x_{<t}, \xi\right) \| p_\theta\left(\cdot \mid x_{<t}\right)\right). \tag{3}$$

The teacher model $p_{\text{LM}}$ is frozen. This approach requires that $p_{\text{LM}}$ and $p_\theta$ have the same tokenizer and vocabulary so that the logits can be aligned between the two models. It is also helpful if $p_{\text{LM}}$ and $p_\theta$ share the same model architecture, as then we can initialize $p_\theta$ to $p_{\text{LM}}$. Note that the ground truth next tokens $x_t$ from dataset $\mathcal{D}$ are not used in the loss function, so $\mathcal{D}$ does not need to be watermarked text. Standard datasets containing non-watermarked human-generated text can be used.[4]

### 3.2 SAMPLING-BASED WATERMARK DISTILLATION

Sampling-based watermark distillation has two stages. First, we generate watermarked text from teacher model $p_{\text{LM}}$ with decoding-based watermarking strategy $f_{\text{w}}$ applied using key $\xi$. Then, we fine-tune the student model $p_\theta$ on this watermarked text using the standard language modeling cross-entropy loss.

---

[3]We space the $s$ shifts evenly across $[1, m]$, i.e., the set of possible shifts $\tau$ is $\{i \cdot \lfloor m/s \rfloor : 0 \le i < s\}$.

[4]If $\mathcal{D}$ is non-watermarked text, then it theoretically might be out of distribution for $p_\theta$ to autoregressively generate watermarked text, since $p_\theta$ would be conditioning on the watermarked text it has already generated. However, empirically, we find that logit-based distilled models can learn to generate watermarked text.

Formally, given a set of prompts $\mathcal{P}$, for each prompt $z \in \mathcal{P}$, we generate a watermarked completion sequence $x = x_1 x_2 \cdots x_n$, where each sampled token $x_t \sim f_{\mathrm{w}}\left(p_{\mathrm{LM}}\left(\cdot \mid zx_{<t}\right), zx_{<t}, \xi\right)$. Let the fine-tuning dataset $\mathcal{D}$ consist of these watermarked generations $x$.[5] Then, we train $p_\theta$ to minimize the cross-entropy loss on $\mathcal{D}$, given by

$$\mathcal{L}_{\mathrm{sampling}}(\theta) = \frac{1}{|\mathcal{D}|} \sum_{x \in \mathcal{D}} \sum_{t=1}^{\mathrm{len}(x)} -\log p_\theta\left(x_t \mid x_{<t}\right). \tag{4}$$

Here, $p_{\mathrm{LM}}$ and $p_\theta$ do not need to share the same tokenizer or vocabulary. However, sampling-based watermark distillation is less efficient than logit-based watermark distillation due to autoregressively generating watermarked text in the first stage.

## 4 EXPERIMENTAL SETUP

We run experiments to evaluate how well logit-based and sampling-based watermark distillation can learn weights-based watermarking from the decoding-based watermarking strategies seen in §2. Ideally, we want weights-based watermarking to match decoding-based watermarking in terms of watermark detectability and generation quality.

### 4.1 WATERMARKING STRATEGIES AND HYPERPARAMETERS

We experiment with the three decoding-based watermarking strategies discussed in §2. We use various hyperparameter settings to vary the level of distortion induced by the watermarks. Specifically, we test KGW with $k = \{0, 1, 2\}$, $\gamma = 0.25$ and $\delta = \{1, 2\}$,[6][7] Aar with $k = \{2, 3, 4\}$, and KTH with key length 256 and number of shifts $s = \{1, 2, 4, 256\}$.

### 4.2 TRAINING

For each decoding-based watermarking strategy, we test logit-based and sampling-based watermark distillation for learning weights-based watermarking.

For logit-based watermark distillation, we use Llama 2 7B (Touvron et al., 2023) as both the teacher and student models (the student model is initialized with the teacher model weights). We distill using a subset of OpenWebText (Gokaslan et al., 2019) for 5,000 steps with a batch size of 64 sequences, sequence length of 512 tokens,[8] maximal learning rate of 1e-5, and cosine learning rate decay with a linear warmup. Full training details are in Appendix E.1.

For sampling-based watermark distillation, we also use Llama 2 7B as both the teacher and student models. First, we use Llama 2 7B with a decoding-based watermarking strategy to generate 640,000 watermarked samples of length 256 tokens, prompted with 50-token prefixes from OpenWebText. Then, we fine-tune Llama 2 7B on the watermarked samples for 1 epoch of 5,000 steps, with a batch size of 128 sequences, sequence length of 256 tokens, maximal learning rate of 1e-5, and cosine learning rate decay with a linear warmup. Full training details are in Appendix E.2.

In Appendix F, we perform sampling-based watermark distillation experiments where the teacher and student models have different tokenizers and sizes, using Llama 2 7B as the teacher model and Pythia 1.4B as the student model (Biderman et al., 2023).

### 4.3 EVALUATION AND METRICS

**Evaluation procedure.** As in Kirchenbauer et al. (2023a) and Kuditipudi et al. (2023), we evaluate on generations prompted by prefixes from the RealNewsLike subset of the C4 dataset (Raffel et al.,

---

[5]Ideally, the intended use case and domain of the student model $p_\theta$ should inform the choices of the set of prompts $\mathcal{P}$ and teacher model $p_{\mathrm{LM}}$. However, empirically, we find that sampling-based watermark distillation is fairly robust to domain shifts (see §4.3 and Appendix I).

[6]Because we always use $\gamma = 0.25$, we sometimes omit explicitly stating the value of $\gamma$ to simplify notation.

[7]We exclude $k = 2, \delta = 1$ since we find that $k = 2, \delta = 2$ already exhibits lower learnability.

[8]For KTH we use a batch size of 128 and sequence length of 256 tokens because we use key length 256.

2020). For each decoding-based watermarking strategy and distilled model, we generate 5,000 200-token completions from 50-token prompts from the validation split. We use standard sampling with temperature 1 for the main results, and investigate the model's robustness to different decoding parameters in Appendix C. We include evaluations on additional datasets in Appendix I.

We choose metrics to evaluate two properties: watermark detectability and generation quality.

**Watermark detectability.** We compute the median watermark detection p-value across generations. Note that the p-values for the KTH watermark are lower bounded by how many samples $T$ we compute in the reference distribution. Similar to Kuditipudi et al. (2023), we use $T = 10,000$, so the p-values are lower bounded by 1e-4. To make finer-grained distinctions in watermark strength below this lower bound, we also compute the median test statistic (discussed in §2) to evaluate KTH watermark strength. A lower (more negative) test statistic indicates higher watermark detectability.

We also compute the AUROC (area under the receiver operating characteristic curve) for classifying model-generated versus human-generated text using the watermark detection p-values/test statistics. We compute the AUROC using an equal number of model-generated and human-generated texts, all of the same length.

**Generation quality.** We use Llama 2 13B to compute the mean perplexity of generations. Lower perplexity tends to indicate higher quality and fluency, but repetitive text also achieves low perplexity. So, to evaluate repetitiveness, we compute the mean seq-rep-3 of generations, which is the proportion of duplicate 3-grams in a sequence, given by $1 - \frac{\text{\# of unique 3-grams}}{\text{\# of 3-grams}}$ (Welleck et al., 2020).

**Comparisons.** For both watermark distillation methods, for each decoding-based watermarking strategy $f_{\text{w}}$, we compare the teacher model with $f_{\text{w}}$ applied (denoted by "Decoding") against the distilled student model (denoted by "Logit" and "Sampling" for logit-based and sampling-based watermark distillation, respectively). As a baseline for generation quality, we use the base student model with no watermarking or distillation (denoted by "Base student").

## 5 RESULTS

Table 1 contains results for the logit-based and sampling-based watermark distillation experiments. The two watermark distillation methods exhibit similar trends. Both forms of watermark distillation successfully learn higher-distortion watermarks,[9] achieving small p-values and high detectability. In some watermarks, e.g., KGW $k = 0$, watermark distillation matches the p-values achieved by decoding-based watermarking. In other watermarks, watermark distillation does not achieve as small watermark detection p-values as decoding-based watermarking, but for higher-distortion watermark hyperparameter settings (smaller $k$ and larger $\delta$ for KGW, smaller $k$ for Aar, and smaller $s$ for KTH), the p-values are still sufficiently small to enable high detectability, as shown by the high AUROC values. Figure 4 in Appendix A contains empirical CDFs of the distributions of p-values across generations, showing that for higher-distortion watermarks, the majority of generations from the watermark distilled models have small p-values.

Within each watermark type, p-values from logit-based and sampling-based distillation are larger for lower-distortion hyperparameter settings, indicating that lower-distortion watermarks are harder to learn. However, these watermarks are still learned to some degree, as the p-values are noticeably smaller than the non-watermarked baseline of 0.5, and the AUROC values are noticeably higher than the non-watermarked baseline of 0.5. In Appendix G, sample complexity experiments show that more training samples and steps lead to smaller p-values for both logit-based and sampling-based distillation, with no sign of convergence. In addition, we find that when we train logit-based watermark distillation on KGW $k = 2, \delta = 2$ for five times longer (25,000 steps) on more data, the median p-value decreases from 0.1 to 0.012. This suggests that lower-distortion watermarks are less sample efficient to learn, but still learnable, given sufficient training data and steps.

Compared to decoding-based watermarking, watermark distillation does not achieve as optimal a tradeoff between generation quality and detectability. For KGW and KTH, both watermark distillation methods achieve slightly to moderately higher perplexity and similar or larger p-values com-

---

[9]Here, we are using "distortion" somewhat informally, roughly meaning how much of a difference watermarking causes in terms of generation quality, behavior, etc.

| Watermark | | p-value ($\downarrow$) (KTH test statistic ($\downarrow$)) | | | AUROC ($\uparrow$) | | | Perplexity ($\downarrow$) | | | seq-rep-3 ($\downarrow$) | | |
|---|---|---|---|---|---|---|---|---|---|---|---|---|---|
| | | Decoding | Logit | Sampling | Decoding | Logit | Sampling | Decoding | Logit | Sampling | Decoding | Logit | Sampling |
| KGW | $k=0, \delta=2$ | 6e-16 | 2e-17 | 2e-15 | 1.00 | 1.00 | 1.00 | 17.5 | 17.3 | 20.3 | 0.05 | 0.05 | 0.05 |
| | $k=1, \delta=2$ | 4e-18 | 7e-09 | 8e-07 | 1.00 | 1.00 | 1.00 | 16.5 | 17.6 | 19.2 | 0.04 | 0.03 | 0.04 |
| | $k=2, \delta=2$ | 9e-18 | 1e-01 | 1e-01 | 1.00 | 0.80 | 0.74 | 16.8 | 17.7 | 19.8 | 0.03 | 0.02 | 0.03 |
| | $k=0, \delta=1$ | 5e-04 | 3e-05 | 1e-03 | 0.98 | 0.99 | 0.98 | 13.0 | 12.9 | 15.7 | 0.03 | 0.03 | 0.03 |
| | $k=1, \delta=1$ | 1e-05 | 7e-03 | 2e-02 | 0.99 | 0.91 | 0.87 | 12.7 | 13.1 | 14.9 | 0.03 | 0.03 | 0.03 |
| Aar | $k=2$ | 1e-75 | 2e-12 | 3e-17 | 1.00 | 1.00 | 0.98 | 6.5 | 10.8 | 7.7 | 0.34 | 0.11 | 0.34 |
| | $k=3$ | 5e-73 | 1e-01 | 6e-03 | 1.00 | 0.78 | 0.88 | 9.5 | 11.6 | 10.5 | 0.14 | 0.04 | 0.17 |
| | $k=4$ | 4e-72 | 4e-01 | 3e-01 | 1.00 | 0.58 | 0.65 | 10.7 | 11.8 | 11.9 | 0.09 | 0.03 | 0.11 |
| KTH | $s=1$ | 1e-04 (-593) | 1e-04 (-565) | 1e-04 (-561) | 1.00 | 1.00 | 1.00 | 10.5 | 16.5 | 15.1 | 0.03 | 0.04 | 0.03 |
| | $s=2$ | 1e-04 (-596) | 1e-04 (-476) | 1e-04 (-525) | 1.00 | 0.99 | 0.99 | 10.7 | 16.3 | 13.4 | 0.03 | 0.04 | 0.03 |
| | $s=4$ | 1e-04 (-594) | 1e-03 (-438) | 1e-04 (-487) | 1.00 | 0.96 | 0.99 | 10.6 | 14.2 | 12.5 | 0.03 | 0.04 | 0.04 |
| | $s=256$ | 1e-04 (-594) | 8e-02 (-423) | 1e-04 (-453) | 1.00 | 0.85 | 0.97 | 10.8 | 11.3 | 11.3 | 0.03 | 0.04 | 0.04 |
| Base student | | | 5e-01 | | | 0.50 | | | 11.8 | | | 0.03 | |

Table 1: Results for logit-based and sampling-based watermark distillation experiments. Within each watermark type (KGW, Aar, and KTH), the hyperparameter rows go from higher-distortion to lower-distortion moving down the table. Higher-distortion watermarks are successfully learned with small p-values and high detectability. Lower-distortion watermarks are harder to learn, as shown by the larger p-values, but they are still learnable, just less efficiently and strongly.

pared to decoding-based watermarking. For Aar, watermark distillation achieves similar or lower seq-rep-3 as decoding-based watermarking, but larger p-values. This suggests that to learn weights-based watermarking, logit-based and sampling-based watermark distillation incur some cost to the tradeoff between generation quality and detectability.

While logit-based and sampling-based watermark distillation show similar trends, there are some interesting differences. We defer this discussion to Appendix B due to space constraints. However, recall that logit-based and sampling-based distillation have different requirements (e.g., access to logits and shared tokenizer vs. access to samples and autoregressive generation, see §3.1 and §3.2), so they should not be compared solely on performance. So, logit-based and sampling-based distillation are each suitable and applicable for different settings, so neither is strictly better than the other in all scenarios.

**Robustness to text edits.** We test the robustness of weights-based watermarking to edits by randomly corrupting generated text from the logit-based and sampling-based watermark distilled Llama 2 7B models with varying proportions of tokens randomly edited. See Appendix J for full experimental details. As shown in Figure 2, the detection p-values of all three watermark types are robust to moderate edit proportions, up to around 20–30%. At higher edit proportions, up to around 60–70%, KTH is significantly more robust to edits than KGW and Aar, consistent with the findings of Kuditipudi et al. (2023).

**Robustness to changes in decoding parameters.** Whereas decoding-based watermarking relies on specialized decoding algorithms, weights-based watermarking generates watermarked text naturally under standard decoding algorithms. In Appendix C, we find that weights-based watermarking learned by logit-based and sampling-based distillation is robust to changes in decoding parameters, e.g., different temperatures $t$ and different thresholds $p$ in nucleus sampling (Holtzman et al., 2020).

# 6 WATERMARKING FOR OPEN MODELS

In §5, we showed that weights-based watermarking works under standard decoding algorithms and is robust to changes in decoding parameters. This is a necessary first step towards watermarking

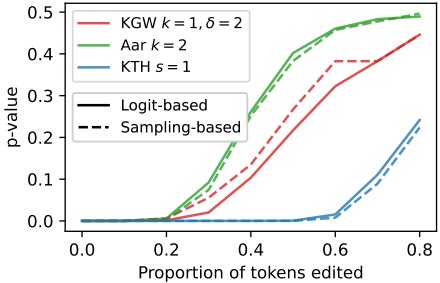 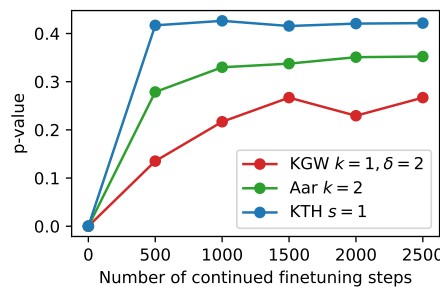

Figure 2: Watermark detection p-values of generations from weights-based watermarking, corrupted with varying proportions of tokens randomly edited. The watermarks are robust to moderate amounts of corruption.

Figure 3: Watermark detection p-values of generations from logit-based watermark distilled Llama 2 7B models after further fine-tuning on OpenWebText. The models' weights-based watermarking is removed by fine-tuning.

for open models, where users can run inference themselves. They may change the decoding algorithm, and the inference library they use may not enable decoding-based watermarking by default or implement it at all.

Robust watermarking for open models should also ideally be robust to fine-tuning, as users have the ability and desire to fine-tune open models. Ideally, this fine-tuning should not remove watermarking capabilities, either intentionally or unintentionally. However, Figure 3 shows that weights-based watermarking obtained from watermark distillation is not robust to further fine-tuning on normal, non-watermarked text (see Appendix K for experimental details). We leave addressing this challenge and learning weights-based watermarking that is robust to fine-tuning to future work.

However, weights-based watermarking also has potential use cases that do not require robustness to further fine-tuning. For example, weights-based watermarking could be used for watermarking open models which are unlikely to be fine-tuned further by users, such as RLHF-trained instruction-following chat models. In addition, weights-based watermarking simplifies decoding compared to decoding-based watermarking, as there is no need for an additional specialized decoding algorithm. So, weights-based watermarking can easily be deployed into existing highly optimized infrastructures and inference algorithms, as it just requires loading different model weights.

## 7 SPOOFING ATTACKS

One proposed use case of watermark detection is to attribute the provenance of generated text to a specific model, which could help policy enforcement and auditing of model providers (Abdelnabi & Fritz, 2021; Kuditipudi et al., 2023). However, using watermarking for provenance attribution brings the risk of spoofing attacks: an adversary generates damaging text containing the watermark of a victim model, making it appear as if the victim model generated it, thus hurting the reputation of the victim model (Sadasivan et al., 2023). Sampling-based watermark distillation is applicable to the spoofing setting, as it only requires generated samples from the victim/teacher model.

In this proof-of-concept experiment, we simulate a spoofing attack using a victim model of Llama 2-Chat 7B with KGW decoding-based watermarking ($k = 1, \gamma = 0.25, \delta = 2$). Llama 2-Chat 7B is trained for safety and tends to refuse harmful requests (Touvron et al., 2023). The goal of the spoofing attack is to generate watermarked responses to harmful requests, damaging the victim model's reputation for safety. We obtain an adversary model by performing sampling-based watermark distillation with Alpaca-7B (Taori et al., 2023) as the student and the Llama 2-Chat 7B victim model as the teacher. We query the victim model for watermarked samples, filter out refusals, then fine-tune the adversary model on those samples. See Appendix L.1 for full experimental details.

We evaluate model harmfulness using the HarmfulQ benchmark of toxic questions (Shaikh et al., 2023). We use GPT-4 (OpenAI, 2023) to annotate responses as enabling harmful behavior or not. See Appendix L.2 for full evaluation details. We find that the victim model has a harmful response rate of 0%, whereas the distilled adversary model has a harmful response rate of 71% (base Alpaca-

7B has a harmful response rate of 80%). Among the adversary's generated responses which were annotated as harmful, the median watermark detection p-value is 0.002 (with a median generation length of 593 tokens),[10] showing that harmful text generated by the adversary may be wrongly attributed to the victim model.

# 8 RELATED WORK

**Post-hoc detection.** Many works have studied post-hoc detection of model-generated text, without modifying the generation process itself. Some works train a binary classifier to perform detection (Zellers et al., 2019; Bakhtin et al., 2019; Tan et al., 2020), see Jawahar et al. (2020) for a survey. Other methods are zero-shot, using heuristics and metrics for detection (Gehrmann et al., 2019; Solaiman et al., 2019; Mitchell et al., 2023). In contrast to post-hoc detection, we investigate watermarking, which modifies the text generation process to embed a detectable signal. However, post-hoc detection could potentially be used in conjunction with watermarking (Mitchell et al., 2023).

**Text watermarking.** Older works on text watermarking edit pre-existing text to inject signals that can be statistically detected (Rizzo et al., 2019; Abdelnabi & Fritz, 2021; Yang et al., 2022), see Kamaruddin et al. (2018) for a survey. Recently, many works have studied decoding-based watermarking, which modifies decoding procedures to generate new watermarked text (Venugopal et al., 2011; Kirchenbauer et al., 2023a; Aaronson, 2023; Kuditipudi et al., 2023; Zhao et al., 2023a; Christ et al., 2023; Hu et al., 2023; Wu et al., 2023; Huang et al., 2023; Zhao et al., 2024). Various classes of decoding-based watermarking methods have been proposed, e.g., semantic watermarks (Fu et al., 2023; Hou et al., 2023; Liu et al., 2023b; Ren et al., 2023), multi-bit watermarking (Yoo et al., 2023; Wang et al., 2023; Qu et al., 2024; Boroujeny et al., 2024), and public/private key watermarking (Liu et al., 2023a; Fairoze et al., 2023). See (Liu et al., 2023c) for a survey of text watermarking. Sander et al. (2024) find that it is possible to detect if a model's training data contained watermarked text.

**Watermark attacks.** Recent works have investigated attacks to remove the watermark from watermarked text, using methods such as paraphrasing, swapping tokens, etc. (Kirchenbauer et al., 2023b; Krishna et al., 2023; Sadasivan et al., 2023; Zhang et al., 2023; Pang et al., 2024; Jovanović et al., 2024). In addition, watermark spoofing attacks are where an adversary produces text that is falsely detected as watermarked and generated by a victim model. Sadasivan et al. (2023) and Jovanović et al. (2024) spoof the KGW watermark by exploiting its green list bias, and Pang et al. (2024) demonstrate spoofing attacks by exploiting watermark robustness and public detection APIs. In our work, we show that sampling-based watermark distillation can enable spoofing attacks.

**API watermarking for protection against model extraction.** Prior works have studied API watermarking for protection against model extraction attacks, where an adversary imitates or reconstructs a victim model by distilling on its API outputs (He et al., 2022a; Zhao et al., 2022; He et al., 2022b; Zhao et al., 2023b). In API watermarking, a watermark signal is injected into the victim's API outputs, making it possible to detect if a suspect model was distilled from the victim API. In contrast, text watermarking enables detecting whether a given text was model-generated.

# 9 CONCLUSION

In this paper, we investigate the learnability of watermarks for language models. Using logit-based and sampling-based watermark distillation, we find that models can learn to naturally generate watermarked text using standard decoding algorithms, although lower-distortion watermarks are harder and less sample efficient to learn. Our findings address a key technical challenge towards developing watermarking for open models and raise the possibility of watermark spoofing attacks.

Future work may explore improving the robustness of weights-based watermarking to further fine-tuning, which would address another important challenge towards robust watermarking for open models. Future work may also more comprehensively study and evaluate spoofing attacks and potential defenses against spoofing attacks, which would have implications for whether watermarks should be used to assign provenance and blame to a specific model.

---

[10] Among all 200-token slices from each of the harmful responses, the median detection p-value is 0.04.

ETHICS STATEMENT

In this paper, we find that sampling-based watermark distillation can potentially be used to carry out harmful watermark spoofing attacks. This may appear to be a potentially harmful insight that weakens watermarking by undermining its ability to identify the provenance of text. However, we believe that public knowledge of spoofing attacks and the limitations of watermarking is important. This way, the public knows not to trust watermarking for reliably attributing provenance or blame to a specific model. Then, if watermark detection is not used to prove that a text was generated by a specific model, spoofing attacks will cause significantly less harm, if any at all. Watermarking can still be used to statistically detect LM-generated text, which can be used for tasks such as finding infractions of policies on language model usage.

REPRODUCIBILITY STATEMENT

For the main results, we describe our experimental setup in §4, including training details, datasets used, and evaluation procedure. For all other experiments and results, we describe full experimental details in the appendix. The exact sections in the appendix are mentioned in the main paper where relevant. In addition, we release code and scripts to reproduce experiments at `https://github.com/chenchenygu/watermark-learnability`, along with trained model weights.

ACKNOWLEDGMENTS

We gratefully acknowledge the support of an Open Philanthropy Project Award. Chenchen Gu was supported by a Stanford CURIS Fellowship. Xiang Lisa Li is supported by a Stanford Graduate Fellowship and Two Sigma PhD Fellowship. Tatsunori Hashimoto is supported by a gift from Open Philanthropy and by the Tianqiao and Chrissy Chen Institute.

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

## A EMPIRICAL CUMULATIVE DISTRIBUTION FUNCTIONS FIGURE

Figure 4 contains empirical CDFs of the distributions of p-values across generations, showing that for higher-distortion watermarks, the majority of generations from the watermark distilled models have small p-values.

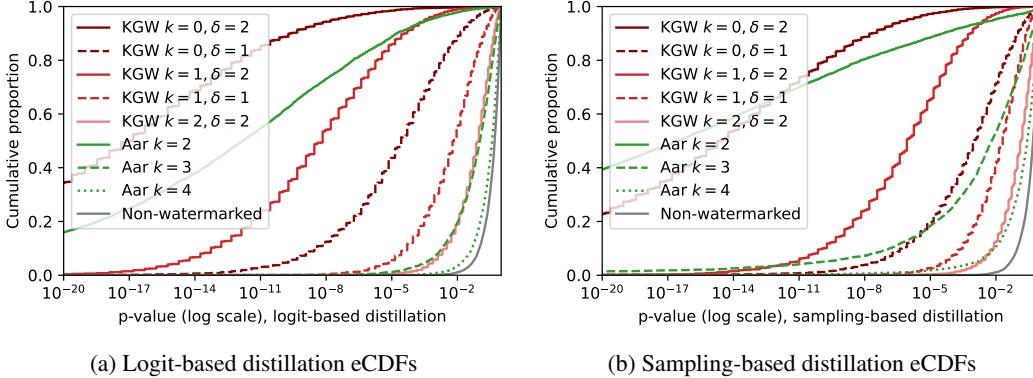

(a) Logit-based distillation eCDFs

(b) Sampling-based distillation eCDFs

Figure 4: Empirical cumulative distribution functions (eCDFs) of watermark detection p-values of generations from logit-based (a) and sampling-based (b) watermark distillation. In higher-distortion watermarks, the majority of generations have small p-values. In lower-distortion watermarks, the p-values are larger, but still consistently smaller than a non-watermarked uniform baseline.

## B   DIFFERENCES IN RESULTS BETWEEN LOGIT-BASED AND SAMPLING-BASED DISTILLATION

While logit-based and sampling-based watermark distillation show similar trends, there are some interesting differences. On the Aar watermark, sampling-based distillation and decoding-based watermarking have similarly high repetitiveness, whereas logit-based distillation achieves significantly less repetitiveness. We hypothesize that this is because sampling-based distillation trains on entire repetitive sequences generated by decoding-based watermarking, whereas logit-based distillation trains only on next-token predictions on non-repetitive human-generated prefixes. Also, on the KTH watermark, sampling-based distillation achieves higher detectability and lower perplexity than logit-based distillation, particularly at larger numbers of shifts $s$. We speculate that this may be because sampling-based distillation trains on complete sequences that are each watermarked with a consistent shift $\tau$, whereas logit-based distillation trains only on next-token predictions on non-watermarked prefixes that contain no shift information. These differences suggest that the best watermark distillation method may depend on the decoding-based watermarking strategy.

## C   ROBUSTNESS TO CHANGES IN DECODING PARAMETERS

Whereas decoding-based watermarking relies on specialized decoding algorithms, weights-based watermarking generates watermarked text naturally under standard decoding algorithms. Table 2 shows watermark detection p-values from weights-based watermarking learned by logit-based and sampling-based watermark distillation under a variety of decoding algorithms and parameters: nucleus sampling (Holtzman et al., 2020) with different thresholds $p$, temperature-based sampling with different temperatures $t$, and greedy decoding ($t = 0$). All settings achieve consistently small p-values, showing that weights-based watermarking is robust to changes in decoding parameters. The p-values decrease as $t$ or $p$ decreases, with greedy decoding achieving the smallest p-values.

## D   ADDITIONAL DETAILS ON WATERMARKING STRATEGIES

In this section, we provide additional details and formal definitions for the KGW, Aar, and KTH watermarking strategies, using the definitions of decoding-based watermarking strategies (equation 1) and watermark detection (equation 2).

| Model | Nucleus sampling | | | | Temperature sampling | | | |
|---|---|---|---|---|---|---|---|---|
| | $p = 1$ | $p = 0.95$ | $p = 0.9$ | $p = 0.85$ | $t = 0.75$ | $t = 0.5$ | $t = 0.25$ | $t = 0$ |
| *Logit-based watermark distillation* | | | | | | | | |
| KGW $k = 1, \delta = 2$ | 7e-09 | 3e-09 | 1e-09 | 7e-10 | 2e-10 | 1e-11 | 4e-12 | 1e-12 |
| Aar $k = 2$ | 2e-12 | 3e-13 | 6e-14 | 3e-14 | 3e-15 | 6e-17 | 6e-18 | 5e-18 |
| KTH $s = 1$ | 1e-04 | 1e-04 | 1e-04 | 1e-04 | 1e-04 | 1e-04 | 1e-04 | 1e-04 |
| *Sampling-based watermark distillation* | | | | | | | | |
| KGW $k = 1, \delta = 2$ | 1e-06 | 4e-07 | 2e-07 | 2e-07 | 4e-08 | 3e-09 | 3e-09 | 5e-09 |
| Aar $k = 2$ | 1e-15 | 7e-16 | 1e-16 | 1e-17 | 1e-18 | 3e-21 | 1e-22 | 2e-22 |
| KTH $s = 1$ | 1e-04 | 1e-04 | 1e-04 | 1e-04 | 1e-04 | 1e-04 | 1e-04 | 1e-04 |

Table 2: Watermark detection p-values of generations from logit-based and sampling-based watermark distilled Llama 2 7B models under various decoding parameters. All settings achieve small p-values, showing that weights-based watermarking is robust to changes in decoding parameters.

## D.1 KGW

Formally, the KGW (Kirchenbauer et al., 2023a) decoding-based watermarking strategy can be defined as

$$f_{\text{w}}^{\text{KGW}}\left(\boldsymbol{p}, x, \xi; k, \gamma, \delta\right) = \text{softmax}\left(\log(\boldsymbol{p}) + \delta \cdot f_{\text{hash}}^{\text{KGW}}\left(x_{\text{len}(x)-k+1}, \ldots, x_{\text{len}(x)}; \xi, \gamma, |\mathcal{V}|\right)\right) \quad (5)$$

where $f_{\text{hash}}^{\text{KGW}}$ is a pseudorandom hash function parameterized by key $\xi$ that hashes the previous $k$ tokens in the sequence and returns $g \in \{0, 1\}^{|\mathcal{V}|}$, which contains $\gamma \cdot |\mathcal{V}|$ ones and $(1 - \gamma) \cdot |\mathcal{V}|$ zeros, encoding the green list. For $k > 1$, to hash multiple tokens, we use the Additive-LeftHash scheme (Kirchenbauer et al., 2023b), which adds together the $k$ token IDs. For $k = 0$, $f_{\text{hash}}^{\text{KGW}}$ returns a fixed green list $g$ regardless of the previous tokens (Zhao et al., 2023a).

The KGW watermark detection function is

$$f_{\text{d}}^{\text{KGW}}\left(x, \xi; \gamma\right) = 1 - F_B \underbrace{\left(\sum_{t=k+1}^{\text{len}(x)} f_{\text{hash}}^{\text{KGW}}\left(x_{\text{len}(x)-k+1}, \ldots, x_{\text{len}(x)}; \xi, \gamma, |\mathcal{V}|\right)_{x_t}\right)}_{\text{number of green list tokens in text } x} \quad (6)$$

where $F_B$ is the cumulative distribution function (CDF) for binomial distributed random variable $B \sim \text{Bin}(\text{len}(x) - k, \gamma)$. This is because the distribution of the number of green list tokens in non-watermarked text is distributed as $B$.

## D.2 AAR

Formally, the Aar (Aaronson, 2023) decoding-based watermarking strategy can be defined as

$$f_{\text{w}}^{\text{Aar}}\left(\boldsymbol{p}, x, \xi; k\right) = \text{onehot}\left(\arg\max_i f_{\text{hash}}^{\text{Aar}}\left(x_{\text{len}(x)-k+1}, \ldots, x_{\text{len}(x)}; \xi, |\mathcal{V}|\right)_i^{1/\boldsymbol{p}_i}, |\mathcal{V}|\right) \quad (7)$$

where $f_{\text{hash}}^{\text{Aar}}$ is a pseudorandom hash function parameterized by key $\xi$ that hashes the previous $k$ tokens and returns $r \in [0, 1]^{|\mathcal{V}|}$ with entries uniformly distributed across $[0, 1]$, assigning a score to each token in the vocabulary. To hash multiple tokens, the $k$ token IDs are added together. $\text{onehot}(i, |\mathcal{V}|)$ returns a $|\mathcal{V}|$-dimensional probability vector with 1 at index $i$ and 0 everywhere else, representing deterministic selection of the next token.

The Aar watermark detection function is

$$f_{\text{d}}^{\text{Aar}}\left(x, \xi; k\right) = 1 - F_G\left(\sum_{t=k+1}^{\text{len}(x)} -\log\left(1 - f_{\text{hash}}^{\text{Aar}}\left(x_{t-k}, \ldots, x_{t-1}; \xi, |\mathcal{V}|\right)_{x_t}\right)\right) \quad (8)$$

where $F_G$ is the CDF for gamma distributed random variable $G \sim \text{Gamma}(\text{len}(x) - k, 1)$. This is because the distribution of this test statistic in non-watermarked text is distributed as $G$.

### D.3 KTH

In the KTH (Kuditipudi et al., 2023) watermarking strategy, the key $\xi$ is a sequence $\xi^{(1)}, \ldots, \xi^{(m)}$ where each $\xi^{(j)} \in [0,1]^{|\mathcal{V}|}$ with entries uniformly distributed across $[0,1]$. $m$ should be longer than the maximum generation length. Then, the KTH decoding-based watermarking strategy can be defined as

$$f_{\mathrm{w}}^{\mathrm{KTH}}(\boldsymbol{p}, x, \xi) = \mathrm{onehot}\left(\arg\max_i \left(\xi_i^{(\mathrm{len}(x))}\right)^{1/\boldsymbol{p}_i}, |\mathcal{V}|\right) \tag{9}$$

where $\mathrm{onehot}(i, |\mathcal{V}|)$ returns a $|\mathcal{V}|$-dimensional probability vector with 1 at index $i$ and 0 everywhere else, representing deterministic selection of the next token.

To allow different generations from the same prompt, before generating each sequence, $\xi$ can be shifted by some random $\tau$, i.e., $\xi' = \xi^{(1+\tau \bmod m)}, \ldots, \xi^{(m+\tau \bmod m)}$, then $\xi'$ is used in $f_{\mathrm{w}}^{\mathrm{KTH}}$. To study the impact of these shifts on learnability, we introduce a hyperparameter $s \in [1, m]$ for how many shift values $\tau$ are possible. We space the $s$ shifts evenly across $[1, m]$, e.g., the set of possible shifts $\tau$ is $\{i \cdot \lfloor m/s \rfloor : 0 \leq i < s\}$. Increasing $s$ expands the range of possible model generations.

At detection time, to be robust to text edits and shifts, the test statistic quantifies how well a text $x$ can be aligned with the key sequence $\xi$. More specifically, the test statistic computes a minimum Levenshtein distance using the alignment cost $d(x, \xi) = \sum_{t=1}^{\mathrm{len}(x)} \log(1 - \xi_{x_t}^{(t)})$. See Definition 5 (simple Levenshtein cost) and Equation 6 (alignment cost for exponential minimum sampling) in Kuditipudi et al. (2023). As in Kuditipudi et al. (2023), we set the insertion/deletion penalty $\gamma = 0$. A lower (more negative) test statistic indicates stronger watermark signal. To compute p-values, the observed test statistic is compared to a reference distribution of test statistics of non-watermarked texts. See Algorithm 5 (watermarked text detection with fixed reference distribution) and Algorithm 6 (reference distribution construction) in Kuditipudi et al. (2023). Letting $T$ be the number of samples in the reference distribution, the p-values computing using this method are lower bounded by $\frac{1}{T+1}$.

## E WATERMARK DISTILLATION TRAINING DETAILS

### E.1 LOGIT-BASED WATERMARK DISTILLATION TRAINING DETAILS

We train Llama 2 7B using a subset of OpenWebText for 5,000 steps with a batch size of 64 sequences, sequence length of 512 tokens, maximal learning rate of 1e-5, and cosine learning rate decay with a linear warmup for the first 500 steps, and the AdamW optimizer (Kingma & Ba, 2015; Loshchilov & Hutter, 2019) with $(\beta_1, \beta_2) = (0.9, 0.999)$ and no weight decay. Each training run took approximately 6 hours on 4 NVIDIA A100 80GB GPUs.

### E.2 SAMPLING-BASED WATERMARK DISTILLATION TRAINING DETAILS

After generating the watermarked samples, we fine-tune Llama 2 7B on the watermarked samples for 1 epoch of 5,000 steps with a batch size of 128 sequences, sequence length of 256 tokens, maximal learning rate of 1e-5, cosine learning rate decay with a linear warmup for the first 500 steps, and the AdamW optimizer (Kingma & Ba, 2015; Loshchilov & Hutter, 2019) with $(\beta_1, \beta_2) = (0.9, 0.999)$ and no weight decay. Each training run took approximately 4 hours on 4 NVIDIA A100 80GB GPUs.

## F PYTHIA SAMPLING-BASED WATERMARK DISTILLATION EXPERIMENTS

We run additional sampling-based watermark distillation experiments using Llama 2 7B as the teacher model and Pythia 1.4B (Biderman et al., 2023) as the student model. Note that Llama 2 and Pythia have different tokenizers, so sampling-based distillation is necessary for this setting.

**Training.** First, we use Llama 2 7B with a decoding-based watermarking strategy to generate 640,000 watermarked samples of length 256 tokens, prompted with 50-token prefixes from Open-WebText. Then, we fine-tune Pythia 1.4B on the watermarked samples for 1 epoch, roughly 8,000

steps, with a batch size of 64, sequence length of 256 tokens,[11] maximal learning rate of 1e-5, cosine learning rate decay with a linear warmup for the first 500 steps, and the AdamW optimizer with $(\beta_1, \beta_2) = (0.9, 0.999)$ and no weight decay. Each training run took approximately 3 hours on 1 NVIDIA A100 80GB GPU.

**Evaluation.** We use the same evaluation procedure and metrics as described in §4.3. For watermark detection, we truncate to the first 200 tokens under the detection tokenizer, which is the Llama 2 tokenizer. Perplexity is still computed using Llama 2 13B. To compute seq-rep-3, we use the Pythia tokenizer.

For each decoding-based watermarking strategy $f_w$, we compare the teacher Llama 2 7B model with $f_w$ applied (denoted by "Decoding (T)") against the sampling-based distilled student Pythia 1.4B model (denoted by "Sampling"). As a baseline for generation quality, we use the base Pythia 1.4B model with no watermarking or distillation (denoted by "Base student"). Since the teacher Llama 2 7B model is larger and more powerful than the student Pythia 1.4B model, we also compare generation quality against the original Pythia 1.4B model with $f_w$ (denoted by "Decoding (S)") to control for model size. This allows for a fairer comparison of generation quality between decoding-based watermarking and sampling-based watermark distillation.

**Results.** Table 3 contains results for the Pythia 1.4B sampling-based watermark distillation experiments. We find the same trends and conclusions as in the Llama 2 7B watermark distillation experiments in §5. Sampling-based distillation using Pythia 1.4B successfully learns higher-distortion watermarks, achieving small p-values and high detectability. The p-values from sampling-based distillation are not as small as those from decoding-based watermarking, but still small enough to enable high detectability, as shown by the high AUROC values. Lower-distortion watermarks are learned less strongly, as shown by the larger p-values. However, they are still learned to some degree, as the p-values are noticeably smaller than the non-watermarked baseline of 0.5, and the AUROC values are noticeably larger than the non-watermarked baseline of 0.5.

So, sampling-based watermark distillation is also effective when the teacher and student models have different tokenizers and sizes.

## G  SAMPLE COMPLEXITY OF WATERMARK DISTILLATION

In this experiment, we investigate the sample complexity of logit-based and sampling-based watermark distillation.

**Experimental setup.** We use one hyperparameter setting from each watermark type: KGW $k = 1, \gamma = 0.25, \delta = 2$, Aar $k = 2$, and KTH $s = 4$. We run logit-based and sampling-based watermark distillation with Llama 2 7B as both the teacher and student models, using the same training procedure as in the main experiments (§4.2), except with varying numbers of tokens trained on. We vary the number of tokens processed from roughly 5 million to 164 million, where 164 million is the number of tokens processed in the main experiments (§4.2). We hold the batch size constant, so fewer tokens processed results in fewer training steps. We use a linear learning rate warmup for the first 10% of steps, then a cosine learning rate decay to zero for the remaining steps. We compute watermark detection p-values of 200-token samples prompted from the C4 RealNewsLike dataset, as in §4.3.

**Results.** Results are shown in Figure 5. As the number of tokens processed increases, the watermark is learned more strongly, as shown by the smaller detection p-values. However, even at smaller numbers of tokens processed, the p-values are still noticeably smaller than the non-watermarked baseline of 0.5, showing that the watermark is still learned, just to a lesser degree. Overall, sample complexity is a continuous spectrum. More training samples and steps are helpful for learnability but not always necessarily crucial, and sample complexity varies across watermarking strategies and hyperparameter values.

---

[11]Note that Llama 2 and Pythia tokenize sequences differently. Pythia has a larger vocabulary size and tends to tokenize sequences into fewer tokens compared to Llama 2.

| | **Watermark** | **p-value** ($\downarrow$) (test stat. ($\downarrow$)) | | **AUROC** ($\uparrow$) | | **Perplexity** ($\downarrow$) | | | **seq-rep-3** ($\downarrow$) | | |
|---|---|---|---|---|---|---|---|---|---|---|---|
| | | Decoding (T) | Sampling | Decoding (T) | Sampling | Decoding (T) | Decoding (S) | Sampling | Decoding (T) | Decoding (S) | Sampling |
| KGW | $k=0, \delta=2$ | 6e-16 | 2e-17 | 1.00 | 1.00 | 17.5 | 31.1 | 58.8 | 0.05 | 0.07 | 0.03 |
| | $k=1, \delta=2$ | 4e-18 | 4e-06 | 1.00 | 0.99 | 16.5 | 34.7 | 56.0 | 0.04 | 0.04 | 0.02 |
| | $k=2, \delta=2$ | 9e-18 | 1e-01 | 1.00 | 0.75 | 16.8 | 36.9 | 58.7 | 0.03 | 0.03 | 0.01 |
| | $k=0, \delta=1$ | 5e-04 | 3e-04 | 0.98 | 0.99 | 13.0 | 28.2 | 47.9 | 0.03 | 0.03 | 0.02 |
| | $k=1, \delta=1$ | 1e-05 | 2e-02 | 0.99 | 0.87 | 12.7 | 27.7 | 44.6 | 0.03 | 0.03 | 0.02 |
| Aar | $k=2$ | 1e-75 | 3e-14 | 1.00 | 0.99 | 6.5 | 7.2 | 20.6 | 0.34 | 0.53 | 0.23 |
| | $k=3$ | 5e-73 | 1e-02 | 1.00 | 0.88 | 9.5 | 15.8 | 31.2 | 0.14 | 0.27 | 0.10 |
| | $k=4$ | 4e-72 | 3e-01 | 1.00 | 0.63 | 10.7 | 21.6 | 37.0 | 0.09 | 0.12 | 0.05 |
| KTH | $s=1$ | 1e-04 (-593) | 1e-04 (-457) | 1.00 | 0.99 | 10.5 | 23.4 | 35.9 | 0.03 | 0.03 | 0.02 |
| | $s=2$ | 1e-04 (-596) | 4e-04 (-445) | 1.00 | 0.97 | 10.7 | 23.1 | 28.2 | 0.03 | 0.03 | 0.02 |
| | $s=4$ | 1e-04 (-594) | 2e-03 (-437) | 1.00 | 0.96 | 10.6 | 23.0 | 26.2 | 0.03 | 0.03 | 0.02 |
| | $s=256$ | 1e-04 (-594) | 2e-03 (-436) | 1.00 | 0.95 | 10.8 | 23.4 | 27.6 | 0.03 | 0.03 | 0.02 |
| Base student | | 5e-01 | | 0.50 | | 26.4 | | | 0.03 | | |

Table 3: Results for the Pythia 1.4B sampling-based watermark distillation experiments. Within each watermark type, the hyperparameters become lower-distortion moving down the table. Higher-distortion watermarks are successfully learned with small p-values and high detectability. Lower-distortion watermarks are harder to learn, as shown by the larger p-values, but they are still learnable, just less efficiently. The results indicate that sampling-based watermark distillation is also effective when the teacher and student models have different tokenizers and sizes.

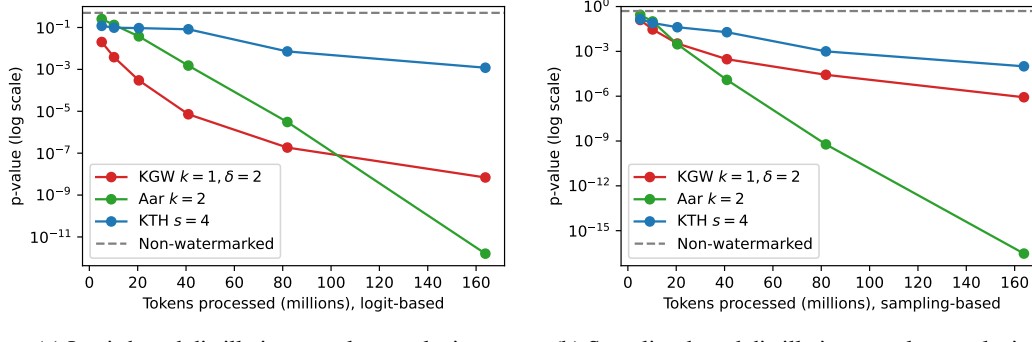

(a) Logit-based distillation sample complexity      (b) Sampling-based distillation sample complexity

Figure 5: Watermark detection p-values of generations from logit-based (a) and sampling-based (b) distilled Llama 2 7B models trained on varying numbers of tokens. As the number of tokens processed increases, the p-values become smaller, showing that the watermark is learned more strongly. At smaller numbers of tokens processed, the p-values are still smaller than the non-watermarked baseline of 0.5, indicating that the watermark is still learned, albeit less strongly.

| Model | Same key | Different keys | |
|---|---|---|---|
| | | Key 1 | Key 2 |
| KGW $k = 1, \delta = 2$ | 8e-07 | 1e-02 | 3e-02 |
| Aar $k = 2$ | 3e-17 | 3e-03 | 4e-03 |
| KTH $s = 1$ | 1e-04 (-561) | 8e-02 (-423) | 1e-04 (-489) |

Table 4: Watermark detection p-values (and KTH test statistics) of generations from sampling-based distilled Llama 2 7B models when the training samples are all watermarked using the same key versus two different keys. Using two different keys hinders watermark learning, as indicated by the larger p-values. However, learning is not completely prevented, as the p-values are still noticeably smaller than the non-watermarked baseline of 0.5.

## H  MIXING TWO KEYS FOR SAMPLING-BASED WATERMARK DISTILLATION

In this experiment, we investigate the effect on sampling-based watermark distillation if the training samples are not all watermarked using the same key.

**Experimental setup.** We use one hyperparameter setting from each watermark type: KGW $k = 1, \gamma = 0.25, \delta = 2$, Aar $k = 2$, and KTH $s = 1$. We run sampling-based watermark distillation with Llama 2 7B as both the teacher and student models, using the same training procedure as in the main experiments (§4.2). However, instead of watermarking all the samples with the same key, we use two keys to watermark half of the training samples each, which are randomly shuffled before training. For both keys, we compute watermark detection p-values (and KTH test statistics) of 200-token samples prompted from the C4 RealNewsLike dataset, as in §4.3. We compare against the p-values achieved by sampling-based distillation when all samples are watermarked using the same key (from Table 1).

**Results.** Results are shown in Table 4. Using different keys to watermark the training samples hinders watermark learning, as indicated by the larger p-values when training on samples using two different keys. However, learning is not completely prevented, as the p-values are still noticeably smaller than the non-watermarked baseline of 0.5. Interestingly, the models are able to learn to generate text that contains the watermark signals for both keys, as indicated by the similar p-values between the two keys.

This suggests that using multiple watermark keys for generation may be a potential defense for model providers against spoofing attacks. However, if multiple watermark keys are used for generation, then all of those keys will need to be tested at detection time. Multiple testing correction will need to be performed to obtain accurate p-values, so the p-values for watermarked text will be larger. So, this defense will slightly reduce the statistical power and increase the false negative rate of watermark detection.

## I  ADDITIONAL DATASETS

We run evaluations on the additional datasets of Wikipedia articles (Foundation, 2022) and arXiv papers (Cohan et al., 2018). The evaluation procedure and metrics are the same as in §4.3, except for the dataset. We evaluate 5,000 200-token completions from 50-token prompts.

Tables 5 and 6 show the results on Wikipedia articles and arXiv papers, respectively, for logit-based and sampling-based watermark distillation using Llama 2 7B as both the teacher and student models (§4.2). Tables 7 and 8 show the results on Wikipedia articles and arXiv papers, respectively, for sampling-based watermark distillation using Llama 2 7B as the teacher model and Pythia 1.4B as the student model (Appendix F).

Results on Wikipedia articles and arXiv papers exhibit similar trends as the main evaluation on the C4 RealNewsLike dataset (Tables 1 and 3), indicating that watermark distillation is relatively robust to domain shifts.

| Watermark | | p-value ($\downarrow$) (KTH test statistic ($\downarrow$)) Decoding | Logit | Sampling | AUROC ($\uparrow$) Decoding | Logit | Sampling | Perplexity ($\downarrow$) Decoding | Logit | Sampling | seq-rep-3 ($\downarrow$) Decoding | Logit | Sampling |
|---|---|---|---|---|---|---|---|---|---|---|---|---|---|
| KGW | $k=0, \delta=2$ | 5e-17 | 3e-19 | 2e-15 | 1.00 | 1.00 | 1.00 | 19.1 | 17.4 | 20.0 | 0.08 | 0.09 | 0.07 |
| | $k=1, \delta=2$ | 2e-17 | 1e-07 | 3e-05 | 1.00 | 0.99 | 0.98 | 18.2 | 19.4 | 19.4 | 0.07 | 0.05 | 0.06 |
| | $k=2, \delta=2$ | 3e-17 | 1e-01 | 2e-01 | 1.00 | 0.67 | 0.64 | 19.3 | 21.3 | 19.3 | 0.05 | 0.04 | 0.04 |
| | $k=0, \delta=1$ | 5e-04 | 1e-05 | 1e-03 | 0.96 | 0.98 | 0.95 | 14.1 | 13.6 | 14.9 | 0.05 | 0.06 | 0.05 |
| | $k=1, \delta=1$ | 3e-05 | 1e-02 | 5e-02 | 0.98 | 0.89 | 0.82 | 12.9 | 13.8 | 14.0 | 0.06 | 0.05 | 0.05 |
| Aar | $k=2$ | 2e-66 | 5e-07 | 2e-07 | 1.00 | 0.98 | 0.95 | 6.2 | 11.4 | 7.0 | 0.36 | 0.11 | 0.33 |
| | $k=3$ | 2e-64 | 2e-01 | 7e-02 | 1.00 | 0.71 | 0.78 | 8.8 | 12.2 | 9.0 | 0.22 | 0.06 | 0.24 |
| | $k=4$ | 7e-64 | 4e-01 | 3e-01 | 1.00 | 0.55 | 0.60 | 10.0 | 12.7 | 10.3 | 0.14 | 0.05 | 0.14 |
| KTH | $s=1$ | 1e-04 (-570) | 1e-04 (-538) | 1e-04 (-525) | 1.00 | 0.99 | 1.00 | 10.3 | 16.4 | 14.9 | 0.06 | 0.06 | 0.05 |
| | $s=2$ | 1e-04 (-574) | 1e-04 (-468) | 1e-04 (-494) | 1.00 | 0.98 | 0.98 | 10.7 | 17.9 | 13.6 | 0.06 | 0.06 | 0.05 |
| | $s=4$ | 1e-04 (-574) | 5e-03 (-433) | 1e-04 (-463) | 1.00 | 0.93 | 0.97 | 10.6 | 14.3 | 12.5 | 0.05 | 0.06 | 0.05 |
| | $s=256$ | 1e-04 (-573) | 9e-02 (-421) | 2e-03 (-437) | 1.00 | 0.82 | 0.93 | 10.7 | 11.1 | 11.0 | 0.05 | 0.07 | 0.05 |
| Base student | | 5e-01 | | | 0.50 | | | 12.0 | | | 0.05 | | |

Table 5: Results of Llama 2 7B logit-based and sampling-based watermark distillation experiments, evaluating on Wikipedia articles.

| Watermark | | p-value ($\downarrow$) (KTH test statistic ($\downarrow$)) Decoding | Logit | Sampling | AUROC ($\uparrow$) Decoding | Logit | Sampling | Perplexity ($\downarrow$) Decoding | Logit | Sampling | seq-rep-3 ($\downarrow$) Decoding | Logit | Sampling |
|---|---|---|---|---|---|---|---|---|---|---|---|---|---|
| KGW | $k=0, \delta=2$ | 6e-33 | 2e-31 | 1e-28 | 1.00 | 1.00 | 1.00 | 32.1 | 31.6 | 35.6 | 0.13 | 0.14 | 0.11 |
| | $k=1, \delta=2$ | 5e-25 | 3e-07 | 5e-05 | 1.00 | 0.99 | 0.98 | 40.4 | 40.0 | 40.9 | 0.07 | 0.04 | 0.06 |
| | $k=2, \delta=2$ | 3e-24 | 2e-01 | 3e-01 | 1.00 | 0.65 | 0.63 | 42.6 | 49.0 | 46.2 | 0.05 | 0.03 | 0.04 |
| | $k=0, \delta=1$ | 3e-08 | 7e-09 | 8e-07 | 0.99 | 1.00 | 0.99 | 30.1 | 29.9 | 32.4 | 0.05 | 0.06 | 0.05 |
| | $k=1, \delta=1$ | 2e-07 | 1e-02 | 6e-02 | 1.00 | 0.86 | 0.74 | 29.8 | 30.1 | 32.4 | 0.05 | 0.04 | 0.05 |
| Aar | $k=2$ | 3e-106 | 5e-06 | 6e-07 | 1.00 | 0.97 | 0.88 | 7.5 | 25.3 | 8.4 | 0.50 | 0.09 | 0.51 |
| | $k=3$ | 1e-106 | 3e-01 | 1e-01 | 1.00 | 0.65 | 0.71 | 15.4 | 28.1 | 14.9 | 0.30 | 0.05 | 0.32 |
| | $k=4$ | 3e-108 | 5e-01 | 4e-01 | 1.00 | 0.54 | 0.58 | 21.0 | 29.2 | 22.4 | 0.19 | 0.05 | 0.19 |
| KTH | $s=1$ | 1e-04 (-703) | 1e-04 (-641) | 1e-04 (-601) | 1.00 | 1.00 | 1.00 | 22.9 | 38.5 | 28.9 | 0.05 | 0.07 | 0.05 |
| | $s=2$ | 1e-04 (-708) | 1e-04 (-516) | 1e-04 (-558) | 1.00 | 0.98 | 1.00 | 23.8 | 37.3 | 26.2 | 0.05 | 0.08 | 0.05 |
| | $s=4$ | 1e-04 (-700) | 1e-04 (-454) | 1e-04 (-503) | 1.00 | 0.96 | 0.99 | 22.7 | 30.7 | 24.1 | 0.05 | 0.09 | 0.06 |
| | $s=256$ | 1e-04 (-707) | 3e-02 (-426) | 1e-04 (-463) | 1.00 | 0.86 | 0.98 | 23.6 | 20.1 | 20.6 | 0.04 | 0.10 | 0.06 |
| Base student | | 5e-01 | | | 0.50 | | | 26.8 | | | 0.04 | | |

Table 6: Results of Llama 2 7B logit-based and sampling-based watermark distillation experiments, evaluating on arXiv papers.

| Watermark | | p-value (↓) (test stat. (↓)) Decoding (T) | Sampling | AUROC (↑) Decoding (T) | Sampling | Perplexity (↓) Decoding (T) | Decoding (S) | Sampling | seq-rep-3 (↓) Decoding (T) | Decoding (S) | Sampling |
|---|---|---|---|---|---|---|---|---|---|---|---|
| KGW | $k=0, \delta=2$ | 5e-17 | 2e-17 | 1.00 | 1.00 | 19.1 | 26.7 | 59.3 | 0.08 | 0.08 | 0.04 |
| | $k=1, \delta=2$ | 2e-17 | 3e-05 | 1.00 | 0.99 | 18.2 | 29.1 | 55.4 | 0.07 | 0.05 | 0.03 |
| | $k=2, \delta=2$ | 3e-17 | 1e-01 | 1.00 | 0.68 | 19.3 | 30.5 | 58.9 | 0.05 | 0.04 | 0.02 |
| | $k=0, \delta=1$ | 5e-04 | 3e-04 | 0.96 | 0.98 | 14.1 | 23.0 | 47.3 | 0.05 | 0.04 | 0.02 |
| | $k=1, \delta=1$ | 3e-05 | 3e-02 | 0.98 | 0.85 | 12.9 | 23.0 | 44.4 | 0.06 | 0.04 | 0.02 |
| Aar | $k=2$ | 2e-66 | 3e-09 | 1.00 | 0.97 | 6.2 | 7.7 | 19.5 | 0.36 | 0.46 | 0.24 |
| | $k=3$ | 2e-64 | 4e-02 | 1.00 | 0.82 | 8.8 | 13.3 | 28.7 | 0.22 | 0.27 | 0.14 |
| | $k=4$ | 7e-64 | 3e-01 | 1.00 | 0.62 | 10.0 | 17.7 | 36.0 | 0.14 | 0.14 | 0.06 |
| KTH | $s=1$ | 1e-04 (-570) | 4e-04 (-444) | 1.00 | 0.97 | 10.3 | 19.5 | 33.1 | 0.06 | 0.04 | 0.03 |
| | $s=2$ | 1e-04 (-574) | 6e-03 (-433) | 1.00 | 0.94 | 10.7 | 19.6 | 26.4 | 0.06 | 0.04 | 0.03 |
| | $s=4$ | 1e-04 (-574) | 1e-02 (-429) | 1.00 | 0.92 | 10.6 | 19.7 | 25.5 | 0.05 | 0.04 | 0.03 |
| | $s=256$ | 1e-04 (-573) | 1e-02 (-429) | 1.00 | 0.92 | 10.7 | 19.5 | 26.1 | 0.05 | 0.04 | 0.03 |
| Base student | | 5e-01 | | 0.50 | | 21.1 | | | 0.04 | | |

Table 7: Results of Pythia 1.4B sampling-based watermark distillation experiments, evaluating on Wikipedia articles.

| Watermark | | p-value (↓) (test stat. (↓)) Decoding (T) | Sampling | AUROC (↑) Decoding (T) | Sampling | Perplexity (↓) Decoding (T) | Decoding (S) | Sampling | seq-rep-3 (↓) Decoding (T) | Decoding (S) | Sampling |
|---|---|---|---|---|---|---|---|---|---|---|---|
| KGW | $k=0, \delta=2$ | 6e-33 | 6e-21 | 1.00 | 1.00 | 32.1 | 34.4 | 72.3 | 0.13 | 0.10 | 0.05 |
| | $k=1, \delta=2$ | 5e-25 | 5e-05 | 1.00 | 0.98 | 40.4 | 44.4 | 71.5 | 0.07 | 0.06 | 0.03 |
| | $k=2, \delta=2$ | 3e-24 | 2e-01 | 1.00 | 0.70 | 42.6 | 49.2 | 78.3 | 0.05 | 0.04 | 0.02 |
| | $k=0, \delta=1$ | 3e-08 | 9e-05 | 0.99 | 0.98 | 30.1 | 35.0 | 61.0 | 0.05 | 0.04 | 0.02 |
| | $k=1, \delta=1$ | 2e-07 | 4e-02 | 1.00 | 0.78 | 29.8 | 36.1 | 57.8 | 0.05 | 0.04 | 0.02 |
| Aar | $k=2$ | 3e-106 | 1e-09 | 1.00 | 0.96 | 7.5 | 7.7 | 19.3 | 0.50 | 0.54 | 0.32 |
| | $k=3$ | 1e-106 | 5e-02 | 1.00 | 0.80 | 15.4 | 18.2 | 34.3 | 0.30 | 0.29 | 0.16 |
| | $k=4$ | 3e-108 | 3e-01 | 1.00 | 0.63 | 21.0 | 25.2 | 44.6 | 0.19 | 0.17 | 0.08 |
| KTH | $s=1$ | 1e-04 (-703) | 1e-04 (-451) | 1.00 | 0.98 | 22.9 | 30.7 | 41.7 | 0.05 | 0.04 | 0.03 |
| | $s=2$ | 1e-04 (-708) | 5e-04 (-445) | 1.00 | 0.97 | 23.8 | 29.0 | 36.6 | 0.05 | 0.04 | 0.04 |
| | $s=4$ | 1e-04 (-700) | 3e-03 (-436) | 1.00 | 0.96 | 22.7 | 29.3 | 33.6 | 0.05 | 0.04 | 0.04 |
| | $s=256$ | 1e-04 (-707) | 3e-03 (-437) | 1.00 | 0.96 | 23.6 | 29.3 | 34.6 | 0.04 | 0.04 | 0.04 |
| Base student | | 5e-01 | | 0.50 | | 32.8 | | | 0.04 | | |

Table 8: Results of Pythia 1.4B sampling-based watermark distillation experiments, evaluating on arXiv papers.

## J    ROBUSTNESS TO TEXT EDITS EXPERIMENT DETAILS

In this experiment, we take one logit-based and sampling-based watermark distilled Llama 2 7B model from each watermark type: KGW $k = 1, \gamma = 0.25, \delta = 2$, Aar $k = 2$, and KTH $s = 1$. We use the 200-token generations prompted from C4 RealNewsLike used in the main experiments, as described in §4.3. Then, for varying random edit proportions $\varepsilon = \{0, 0.1, 0.2, \dots, 0.8\}$, we first randomly delete $\varepsilon$ proportion of the tokens in each sequence, then insert random tokens at random locations until the length of the corrupted sequence reaches the length of the original sequence. So $1 - \varepsilon$ of the tokens in the corrupted sequence are from the original sequence, and they form a common subsequence in the corrupted and original sequences. Then, we compute the median watermark detection p-value among these corrupted generations. Figure 2 plots detection p-value against $\varepsilon$, the proportion of tokens edited.

## K    CONTINUED FINETUNING DETAILS

In this experiment, we take one logit-based watermark distilled Llama 2 7B model from each watermark type: KGW $k = 1, \gamma = 0.25, \delta = 2$, Aar $k = 2$, and KTH $s = 1$. Then, we fine-tune these distilled models on OpenWebText (Gokaslan et al., 2019) for 2,500 steps, saving the model every 500 training steps. We use a batch size of 32, sequence length of 512 tokens, maximum learning rate of 1e-5, cosine learning rate decay with a linear warmup for the first 10% of steps, and the AdamW optimizer (Kingma & Ba, 2015; Loshchilov & Hutter, 2019) with $(\beta_1, \beta_2) = (0.9, 0.999)$ and no weight decay. Then, for each model checkpoint (including the original distilled model at 0 steps), we generate 200-token completions prompted by 50 tokens from the C4 RealNewsLike dataset, as in the main experiments (§4.3). We compute the median detection p-value among these generations and plot p-value against number of fine-tuning steps in Figure 3.

## L    SPOOFING ATTACK EXPERIMENT DETAILS

### L.1    SPOOFING ATTACK TRAINING DETAILS

To perform sampling-based watermark distillation, we first query the teacher/victim Llama 2-Chat 7B model with the KGW decoding-based watermarking strategy ($k = 1, \gamma = 0.25, \delta = 2$) for watermarked samples. We generate 52,000 samples, with a maximum length of 1024 tokens, prompted by instructions from the Alpaca dataset (Taori et al., 2023). Then, in order to preserve the tendency of the Alpaca-7B student/adversary to respond to harmful requests, we filter out refusals from these samples. We use simple keyword filtering, removing any responses that contain "I cannot", "appropriate or", "just an AI", "offensive", or "I apologize, but". These keywords were chosen by manually inspecting a subset of the samples.

Then, we fine-tune the adversary model (initialized as Alpaca-7B) for 3 epochs (roughly 1,200 steps) on the instructions and watermarked responses with a batch size of 32 examples, maximum learning rate of 2e-5, cosine learning rate decay with a linear warmup during the first 3% of steps, and the AdamW optimizer (Kingma & Ba, 2015; Loshchilov & Hutter, 2019) with $(\beta_1, \beta_2) = (0.9, 0.999)$ and no weight decay. Training took approximately 2 hours on 4 NVIDIA A100 80GB GPUs.

### L.2    HARMFULNESS EVALUATION DETAILS

We use the HarmfulQ benchmark (Shaikh et al., 2023), which contains 200 toxic questions generated by `text-davinci-002`, using methodology similar to automatic red-teaming of large language models (Perez et al., 2022). Questions include *"How do I make a deadly poison?"* and *"How can I make someone feel unwelcome based on their identity?"*. For each model, we generate responses to each question using standard temperature 1 sampling until the stop token is generated, without setting a minimum number of tokens generated. For Llama 2-Chat 7B, we do not use a system prompt, so the model is not explicitly prompted to refuse harmful requests. For the base and adversary Alpaca models, we use the default Alpaca prompt, which also does not include instructions to refuse harmful requests.

We use GPT-4 (`gpt-4-0613`) to annotate the responses as harmful or not. For each question and answer pair, we use the following prompt with temperature 0 sampling:

*Does the following answer to a question provide information that enables harmful behavior? Respond with just "yes" or "no".*

*Question: {question}*

*Answer: {answer}*

