# OpenReview forum: "On the Learnability of Watermarks for Language Models"
_ICLR.cc/2024/Conference — ICLR 2024 poster_

### Official Review · Reviewer_GhKC · 2023-10-28

**Soundness:** 3 good
**Presentation:** 2 fair
**Contribution:** 3 good
**Rating:** 6
**Confidence:** 4

**Summary:**

This paper investigates the learnability of watermarks for language models. The authors use watermark distillation, a method that trains a student model to behave like a teacher model that uses decoding-based watermarking. They test their approach on three decoding-based watermarking strategies and find that models can learn to generate watermarked text with high detectability. However, they also find limitations to learnability, such as the loss of watermarking capabilities under fine-tuning on normal text and high sample complexity when learning low-distortion watermarks.

**Strengths:**

1. This paper finds an approach for learning watermarks in language models through knowledge distillation.
2. The paper presents empirical findings on the learnability of watermarks, including the success of the proposed distillation method on three distinct decoding-based watermarking strategies. The results show that models can learn to generate watermarked text with high detectability.
3. The paper discusses the limitations of learnability, including the loss of watermarking capabilities under fine-tuning and the high sample complexity for learning low-distortion watermarks. These limitations provide valuable insights and highlight the challenges in implementing watermarking in language models.

**Weaknesses:**

1. The method presented is neither novel nor new. Knowledge distillation is a well-established concept, and this paper simply tests existing watermarking methods in conjunction with it. The paper appears more evaluative in nature.

2. There's a body of prior works focused on watermark distillation. These works show that if the student model distills the watermarked teacher model, it offers protection against model extraction attacks. Although the context may differ, it is better for the author to acknowledge this in the related works section. Key references include:

- He et al., "Protecting Intellectual Property of Language Generation APIs with Lexical Watermark," AAAI 2022.
- Zhao et al., "Distillation-Resistant Watermarking for Model Protection in NLP," EMNLP 2022.
- He et al., "CATER: Intellectual Property Protection on Text Generation APIs via Conditional Watermarks," NeurIPS 2022.
- Zhao et al., "Protecting Language Generation Models via Invisible Watermarking," ICML 2023.

3. Although the p-value is utilized for watermark detection, I also recommend that the author present quantitative metrics such as AUC, FPR, etc., for a more comprehensive understanding of watermark detection.

**Questions:**

None

---

> ### Author Response · Authors · 2023-11-21
> **Reply to Reviewer GhKC**
>
> We thank you for the useful and insightful feedback and for taking the time to review our paper. We address questions and concerns below.
>
> **Contribution**
>
> Please see our general comment clarifying our contribution. https://openreview.net/forum?id=9k0krNzvlV&noteId=UyaqBwxSNh Our main contribution is our empirical investigation and findings on watermark learnability. It is true that knowledge distillation is a well-established concept, but our method is not our primary contribution. Our findings and evaluations are novel and new, as the learnability of decoding-based text watermarking was previously unexplored, and it was unknown whether existing decoding-based watermarks are learnable or not.
>
> **Relationship to model watermarking for protection against model extraction attacks**
>
> We have updated the related work section to include a discussion of the relationship of these provided works with ours.
>
> These works concern model watermarking for protection against model extraction attacks. Model watermarking for model protection is crucially different from text watermarking, which we study. Model watermarking is meant to protect language model APIs from model stealing/extraction attacks, where an adversary trains a model on outputs from the victim model API. The model watermark enables detecting if a suspect model was extracted from the victim model. Detection is done by querying the suspect model multiple times on specially chosen inputs. The model watermark is specifically designed to be learnable, so that the adversary model learns the watermark during the model extraction attack. On the other hand, text watermarking enables detecting if a given text was model-generated. Detection does not need access to any language model, only the tokenizer, and detection is meant to be performed on any generated text, regardless of the input prompt. Existing decoding-based text watermarking methods are specifically designed to be subtle and imperceptible in order to have minimal impact on text quality, and it was previously unknown whether they are learnable or not.
>
> In short, the object being detected is different (model itself versus generated text), and model watermarking for model protection is specifically designed to be learnable, whereas the learnability of existing text watermarks was previously unexplored and unknown. Because of these crucial differences, these prior works into model watermarking for protection against model extraction attacks do not diminish the value and significance of our findings.
>
>
> **AUROC and FPR**
>
> As requested, we have included the AUROC metric for classifying watermarked and non-watermarked text for a more comprehensive understanding of watermark detection, as shown below for the main logits-based watermark distillation (WD-L) and sampling-based watermark distillation (WD-S) experiments. Dec indicates decoding-based watermarking. Additional full tables can be found in the paper.
>
> | Model | Dec AUROC |WD-L AUROC | WD-S AUROC |
> | --- | --- | --- | --- |
> | KGW $\delta=2$ | 1.000 | 0.997 | 0.995 |
> | KGW $\delta=1$ | 0.992 | 0.913 | 0.874 |
> | Aar $k=2$ | 0.999 | 0.996 | 0.993 |
> | Aar $k=3$ | 1.000 | 0.773 | 0.866 |
> | Aar $k=4$ | 0.999 | 0.580 | 0.631 |
> | KTH $s=1$ | 0.994 | 0.998 | 0.989 |
> | KTH $s=2$ | 0.998 | 0.994 | 0.972 |
> | KTH $s=4$ | 0.995 | 0.969 | 0.955 |
> | KTH $s=256$ | 0.996 | 0.893 | 0.955 |
>
> Because watermark detection returns p-values, the false positive rate is equal to the significance level that is chosen as the threshold for classifying a text as model-generated or not. So we do not report FPR because it is set by the detection user depending on the FPR they are willing to tolerate.

---

> ### Author Response · Authors · 2023-11-22
> **End of discussion period**
>
> Dear Reviewer GhKC,
>
>
> We appreciate all of the valuable time and effort you have spent reviewing our paper. As today is the last day of the discussion period, we gently request that you review our reply and consider updating your evaluation accordingly. We believe that we have addressed all questions and concerns raised, but please feel free to ask any clarifying questions you might have before the end of the discussion period.
>
>
> Best,
>
>
> Authors

---

### Official Review · Reviewer_hxrQ · 2023-10-30

**Soundness:** 3 good
**Presentation:** 2 fair
**Contribution:** 3 good
**Rating:** 6
**Confidence:** 4

**Summary:**

The paper investigates whether watermarks for language models, which allow for the detecting model-generated text, can be directly learned by another model without applying the same watermarking decoding algorithm, termed weights-based watermarking. This has implications for using watermarks in open-source models and for potential spoofing attacks.

The authors propose two methods for learning weights-based watermarks without specialized decoding - logits-based and sampling-based watermark distillation. This transfers watermarks from a teacher model to a student model.

Experiments are run with three different decoding-based watermarking techniques. The results show that the proposed distillation methods can learn watermarks, but learnability varies across schemes and hyperparameters. Lower distortion watermarks are harder to learn.

The authors demonstrate two applications of the findings. For open models, learned watermarks will be removed by further fine-tuning, and they leave the learning weights-based watermarking that is robust to fine-tuning for future work. For spoofing attacks, sampling-based distillation can replicate watermarks. This suggests limitations of using watermarks for provenance.

In summary, this paper provides an interesting empirical analysis of the learnability of current decoding-based textual watermarks. It highlights the potential limitations of text watermarks for provenance. Their findings suggest the need for investigation on watermarks that are difficult to learn as a prevention of the discussed spoofing attacks.

**Strengths:**

Originality:
- Investigates the novel problem of whether language models can learn to generate watermarks themselves. This question has important implications but was previously unexplored.

Quality:
- Provides extensive empirical results across three watermarking schemes and hyperparameters—thorough, reproducible experiments.
- Uses appropriate metrics to assess watermark detection and text quality—rigorous quantitative evaluation.
- Clear methodology and training details—enable replicability.

Clarity:
- Structured logically to build understanding incrementally. Smooth flow between sections.

Significance:
- Foundational insights into watermark learnability advance knowledge in an important emerging area.
- Learnability findings have direct implications for applications like watermarking open models.
- Identifies risks around spoofing that impact watermark security. Informs responsible development.
- Benchmark results across schemes and hyperparameters enable future work. Valuable analysis.

**Weaknesses:**

The experimental methodology could be strengthened by
 1. controlling for model architecture (use the Llama-2 as the student model for both logits-based and sample-based experiments for a more intuitive comparison of the results)). This would better isolate the distillation techniques themselves.
 2. increasing the dataset diversity (currently, only one dataset is used).
 3. somehow, seq-rep-3 is not evaluated on the sample-based distillation.

The technical writing is condensed in some areas, making it harder to follow for non-experts. Expanding explanations around key concepts like the distillation objectives could improve readability.

The related work could be expanded to provide a broader context. Watermarking has been studied for other modalities like images, which could give a useful perspective. Meanwhile, model-specific watermark leveraging the learnability of distribution difference has also been discussed in the literature [He, Xuanli, et al. ].

The spoofing attack evaluation is limited to a simple harmful/innocuous categorization. More nuanced human evaluations could better characterize risks.

The limitations around fine-tuning robustness are acknowledged but not thoroughly investigated. Exploring techniques to improve robustness could be valuable.

He, Xuanli, et al. "Cater: Intellectual property protection on text generation apis via conditional watermarks." Advances in Neural Information Processing Systems 35 (2022): 5431-5445.

**Questions:**

I would like to request an evaluation using Llama-2 for both distillation techniques, rather than different models for each. Doing so would allow for a more direct comparison of the distillation methods themselves, better isolating the impact of the techniques. This could strengthen the technical contributions by providing improved analysis of the relative efficacy of logits-based vs sampling-based distillation.

Additionally, conducting more in-depth studies on spoofing attacks across diverse topics and datasets could further increase the impact of the work's findings. Expanding the spoofing experiments to include varied contexts beyond the current proof of concept could substantiate the risks posed. Rigorously demonstrating successful spoofing in different settings would validate concerns around watermark security and attribution. This expanded analysis could strengthen the evaluative assessment by demonstrating comprehensive consideration of the practical implications.

---

> ### Author Response · Authors · 2023-11-21
> **Reply to Reviewer hxrQ (part 1/2)**
>
> We thank you for the useful and insightful feedback and for taking the time to review our paper. We are glad that you found our findings valuable and significant, and our experiments thorough and rigorous. We address questions and concerns below.
>
> **Llama-2 sampling-based watermark distillation evaluation**
>
> As requested, we have included an evaluation of sampling-based watermark distillation using Llama 2 7B as both the teacher and student model. We use the same evaluation procedure as before. Results are shown below, and also have been added to the paper. The trends are broadly similar to the original logits-based and sampling-based distillation experiments.
>
> | Model | Dec p-value | WD-S p-value | Dec AUROC | WD-S AUROC | Dec LM-score | WD-S LM-score | Dec seq-rep-3 | WD-S seq-rep-3 | Dec MAUVE | WD-S MAUVE |
> | --- | --- | --- | --- | --- | --- | --- | --- | --- | --- | --- |
> | KGW $\delta=2$ | 4.4e-18 | 8.5e-07 | 1.000 | 0.996 | 16.4 | 19.7 | 0.04 | 0.04 | 0.89 | 0.85 |
> | KGW $\delta=1$ | 1.0e-05 | 2.0e-02 | 0.992 | 0.869 | 12.5 | 15.4 | 0.03 | 0.03 | 0.91 | 0.89 |
> | Aar $k=2$ | 1.3e-75 | 5.1e-17 | 0.999 | 0.984 | 6.5 | 7.6 | 0.34 | 0.34 | 0.28 | 0.29 |
> | Aar $k=3$ | 4.8e-73 | 8.5e-03 | 1.000 | 0.873 | 9.5 | 10.5 | 0.14 | 0.17 | 0.74 | 0.66 |
> | Aar $k=4$ | 4.0e-72 | 2.6e-01 | 0.999 | 0.651 | 10.7 | 11.9 | 0.09 | 0.10 | 0.86 | 0.83 |
> | KTH $s=1$ | 1.0e-04 (596) | 1.0e-04 (560) | 0.994 | 0.998 | 10.5 | 15.3 | 0.03 | 0.03 | 0.88 | 0.86 |
> | KTH $s=2$ | 1.0e-04 (600) | 1.0e-04 (526) | 0.998 | 0.994 | 10.7 | 13.5 | 0.03 | 0.04 | 0.91 | 0.87 |
> | KTH $s=4$ | 1.0e-04 (593) | 1.0e-04 (485) | 0.995 | 0.984 | 10.5 | 12.6 | 0.03 | 0.04 | 0.91 | 0.87 |
> | KTH $s=256$ | 1.0e-04 (600) | 1.0e-04 (455) | 0.996 | 0.968 | 10.61 | 11.3 | 0.03 | 0.04 | 0.92 | 0.90 |
> |Base student | 5.0e-01 | | 0.500 | | 11.7 | | 0.03 | | 0.92 | |
>
> However, we note that logits-based and sampling-based distillation have key differences, which means that they should not be compared solely based on efficacy to determine which method is better. Logits-based distillation requires white-box access to the teacher model and that the teacher and student models share the same tokenizer and vocabulary. It is also convenient if the teacher and student models share the same model architecture. On the other hand, sampling-based distillation only requires black-box access to samples from the teacher model, and the teacher and student models can have different tokenizers and vocabularies. Also, sampling-based distillation requires autoregressive sampling from the teacher model, which is slow, whereas logits-based distillation does not require sampling and is faster overall. So, logits-based and sampling-based distillation are each suitable and applicable for different settings, so neither is strictly better than the other across all scenarios.
>
> **Increasing dataset diversity**
>
> We have included additional evaluations of logits-based and sampling-based watermark distillation (including both Llama 2 7B and Pythia 1.4B as student model) on two additional datasets, Wikipedia articles and arXiv papers. We observe similar trends to the original main evaluation on the C4 RealNewsLike dataset. The full results are in Appendix F.
>
> **Seq-rep-3 on sampling-based distillation**
>
> Apologies for the confusion, we originally omitted seq-rep-3 on sampling-based distillation in order to respect margin constraints, but we have now re-included it in Appendix B, as shown below.
>
> | Model | DecT seq-rep-3 | DecS seq-rep-3 | WD-S seq-rep-3 |
> | --- | --- | --- | --- |
> | KGW $\delta=2$ | 0.04 | 0.04 | 0.02 |
> | KGW $\delta=1$ | 0.03 | 0.03 | 0.02 |
> | Aar $k=2$ | 0.34 | 0.53 | 0.23 |
> | Aar $k=3$ | 0.14 | 0.27 | 0.10 |
> | Aar $k=4$ | 0.09 | 0.12 | 0.05 |
> | KTH $s=1$ | 0.03 | 0.03 | 0.02 |
> | KTH $s=2$ | 0.03 | 0.03 | 0.02 |
> | KTH $s=4$ | 0.03 | 0.03 | 0.02 |
> | KTH $s=256$ | 0.03 | 0.03 | 0.02 |
> |Base student | | 0.03 | |
>
> **Expanding explanations around distillation objectives**
>
> We have expanded the explanations around the watermark distillation objectives to improve readability.

---

> ### Author Response · Authors · 2023-11-21
> **Reply to Reviewer hxrQ (part 2/2)**
>
> **Related work relationship to model watermarking**
>
> We have expanded the related work section to include a discussion of model watermarking methods for protection against model extraction attacks [1, 2, 3, 4]. Note that text watermarking, which we study, is crucially different from model watermarking/protection in both setting and method. Model watermarking is meant to protect language model APIs from model stealing/extraction attacks, where an adversary trains a model on outputs from the victim model API. The model watermark enables detecting if a suspect model was extracted from the victim model. Detection is done by querying the suspect model multiple times on specially chosen inputs. The model watermark is specifically designed to be learnable, so that the adversary model learns the watermark during the model extraction attack. On the other hand, text watermarking enables detecting if a given text was model-generated. Detection does not need access to any language model, only the tokenizer, and detection is meant to be performed on any generated text, regardless of the input prompt. Existing decoding-based text watermarking methods are specifically designed to be subtle and imperceptible in order to have minimal impact on text quality, and it was previously unknown whether they are learnable or not.
>
> 1. He et al., "Protecting Intellectual Property of Language Generation APIs with Lexical Watermark," AAAI 2022.
> 2. Zhao et al., "Distillation-Resistant Watermarking for Model Protection in NLP," EMNLP 2022.
> 3. He et al., "CATER: Intellectual Property Protection on Text Generation APIs via Conditional Watermarks," NeurIPS 2022.
> 4. Zhao et al., "Protecting Language Generation Models via Invisible Watermarking," ICML 2023.
>
>
> **Robustness to fine-tuning**
>
> We agree that improving robustness of weights-based watermarking to fine-tuning is an important challenge towards developing watermarking for open models. However, we believe that methods to improve robustness to fine-tuning are not as directly related to our main focus of investigating watermark learnability and would best be explored in future work.
>
> **More in-depth spoofing experiments**
>
> We agree that more in-depth studies and evaluations of watermark spoofing attacks across different settings would be valuable for more comprehensively evaluating the practical implications and risks of spoofing attacks. However, we believe that our findings are still valuable and lay a foundation for future work in this direction. We demonstrate that sampling-based distillation is able to learn to generate watermarked text from black-box watermarked samples, which is the setting for spoofing attacks. We also include experiments on sample efficiency (Appendix D) and different generation datasets (Appendix F). These findings suggest the possibility of spoofing attacks, and we show a simple proof-of-concept spoofing attack to substantiate their possibility. We hope that our work will inform future research and responsible usage of watermark detection.

---

> > ### Comment · Reviewer_hxrQ · 2023-11-21
> > **Score Changed**
> >
> > Thank you for the detailed response and additional results. My concerns are partially resolved. The findings are interesting, though not quite comprehensive or novel enough in my view to warrant an outstanding score. I am raising my score to a 6 to reflect the contribution made.

---

### Official Review · Reviewer_metd · 2023-10-30

**Soundness:** 4 excellent
**Presentation:** 4 excellent
**Contribution:** 2 fair
**Rating:** 6
**Confidence:** 4

**Summary:**

The authors study whether language watermarks can be spoofed by training a student language model (LM) on the watermarked outputs of a teacher LM (the process is known as distillation). The experiments show that all three surveyed watermarking methods can be spoofed when the attacker has 640k (watermarked) samples, each comprised of 256 tokens, and that the distilled watermark remains effective under different decoding strategies and has similar robustness properties as the teacher's watermark.

**Strengths:**

* Overall, I liked the paper. The problem of watermarking open models and spoofing attacks against watermarks is timely and important.

* The authors show that three watermarking methods are vulnerable to spoofing through distillation.

* The authors convince me their distillation approach works for many different generation parameters against all three watermarking methods when sufficiently many samples are available to the attacker.

* The authors show experiments with many state-of-the-art models.

**Weaknesses:**

**Contribution to Open Model Watermarking**. As the authors show, open-model watermarking using distillation is not robust against fine-tuning. I am unclear about the contribution of the authors to open model watermarking. No prior work has used distillation to watermark LMs, hence there is no security threat. What is the use of a watermark that (i) lacks robustness and (ii) can be spoofed by design? I would love to hear the author's thoughts on this.

**Limited novelty.** One would expect that watermarking can be spoofed if the attacker can access sufficiently many watermarked samples. If I understood correctly, the authors use **163 million tokens** ($640$k samples times 256 tokens each) of watermarked text with the _same_ message. How realistic is this assumption? Is the attack effective when a provider uses a different message after generating a set number of tokens?

Spoofing is possible when sufficiently many samples are revealed. I believe a more interesting approach is to measure the number of samples that enable spoofing or to derive a theoretical bound for the spoofer's sample efficiency. Insights into that question could guide the party that injects the watermark into deploying defenses, such as switching to a different key or message after this number of samples has been revealed. Unfortunately, the paper does not investigate this problem. Do the authors have any insights into that problem?

**Questions:**

* What are your contributions towards open model watermarking?

* How sample-efficient is the spoofing attack?

* Can the distilled and original watermarked distributions be distinguished? I assume the spoofed watermarked does not learn rare watermarked sequences, but instead focuses first on the most often occurring ones. If a defender can detect and ignore those during detection, can they evade your spoofing attack while effectively detecting watermarked text?

**Details Of Ethics Concerns:**

I believe there are no ethical concerns in this paper.

---

> ### Author Response · Authors · 2023-11-21
> **Reply to Reviewer metd (part 1/2)**
>
> We thank you for the useful feedback and for taking the time to review our paper. We are glad that you liked the paper and think that the problem is timely and important. We address questions and concerns below.
>
> **Contribution to open model watermarking**
>
> Please see our general comment clarifying our contribution. https://openreview.net/forum?id=9k0krNzvlV&noteId=UyaqBwxSNh One important technical challenge towards open model watermarking is robustness to decoding procedure, since users of open models can run inference themselves and choose the decoding procedure, so it is easy to not use a decoding-based watermark algorithm. Our findings on watermark learnability and weights-based watermarking address this key challenge, taking a necessary first step towards developing open model watermarking. However, as you correctly point out, robustness to fine-tuning is another important technical challenge which we leave for future work. We believe that our contribution is still valuable because it investigates a necessary first step towards open model watermarking that was previously unexplored.
>
> Regarding point (ii), we do not design watermarks to be spoofable. We investigate the learnability of existing decoding-based watermarks, rather than design new watermark algorithms, and high learnability suggests the possibility of being vulnerable to spoofing attacks. Spoofability suggests that watermarks should not be used to assign provenance or blame to a specific model, but it does not hinder the primary case of detecting model-generated text, e.g., preventing students from cheating on essays or preventing LLM-generated misinformation campaigns. Watermarking is still useful and valuable even if it is spoofable.
>
> **Sample efficiency**
>
> To investigate sample efficiency, we have included additional experiments measuring watermark detection p-values as we vary the number of training samples for sampling-based watermark distillation. As in the main experiments, each sample is 256 tokens long, and we use Llama 2 7B as the teacher model and Pythia 1.4B as the student model. Note that the training batch size is held the same, so fewer training samples means fewer gradient steps. P-value results are shown below, and we have also included a plot of p-values against the number of training samples in the paper (Appendix D).
>
> 		               Number of training samples
> | Model | 20k | 40k | 80k | 160k | 320k | 640k |
> | --- | --- | --- | --- | --- | --- | --- |
> | KGW $\delta = 2$| 4.1e-02 | 1.4e-02 | 2.4e-03 | 5.2e-04 | 5.1e-05 | 3.6e-06 |
> | Aar $k = 2$ | 7.4e-02 | 9.4e-03 | 3.5e-04 | 1.0e-06 | 5.1e-10 | 2.9e-14 |
> | KTH $s = 1$ | 7.4e-02 | 4.2e-02 | 1.0e-02 | 1.1e-03 | 6.0e-04 | 1.0e-04 |
>
>
> As expected, the results show that as the number of training samples increases, the watermark is learned more strongly. However, even at small numbers of training samples, the watermark is still learned (the p-values are much lower than chance), just to a lesser degree. Overall, we note that watermark learnability and sample efficiency is a continuous spectrum. More samples are helpful but not necessarily always crucial, and sample efficiency varies across watermarking strategies and hyperparameters.
>
>
> **Using multiple different keys/messages**
>
> We run additional experiments in Appendix E to investigate the behavior of sampling-based watermark distillation if the samples are watermarked using multiple different keys/messages. We also vary the number of training samples as before. The table below shows watermark detection p-values.
>
> 		                     Number of training samples
> | Model | 20k | 40k | 80k | 160k | 320k | 640k |
> | --- | --- | --- | --- | --- | --- | --- |
> | KGW $\delta = 2$ (one key) | 4.1e-02 | 1.4e-02 | 2.4e-03 | 5.2e-04 | 5.1e-05 | 3.6e-06 |
> | KGW $\delta = 2$ (two keys) | 1.7e-01 | 1.4e-01 | 7.8e-02 | 5.7e-02 | 2.9e-02 | 1.4e-02 |
> | Aar $k = 2$ (one key) | 7.4e-02 | 9.4e-03 | 3.5e-04 | 1.0e-06 | 5.1e-10 | 2.9e-14 |
> | Aar $k = 2$ (two keys) | 4.2e-01 | 2.3e-01 | 1.2e-01 | 4.0e-02 | 1.0e-02 | 1.2e-03 |
>
> The results show that using two different keys hinders watermark learning, as indicated by the larger p-values. However, learning is not completely prevented, as the p-values are still much lower than 0.5 and decrease as the number of training samples increases.
>
> This suggests that using multiple watermark keys for generation may be a potential defense against spoofing attacks. However, if multiple watermark keys are used for generation, then all of those keys will need to be tested at detection, so multiple testing correction will need to be performed to obtain accurate, weaker p-values. So, this defense will slightly reduce the statistical power and increase the false negative rate of watermark detection.

---

> ### Author Response · Authors · 2023-11-21
> **Reply to Reviewer metd (part 2/2)**
>
> **Distinguishing between distilled and original watermarked distributions**
>
> Your assumption that the spoofed watermark tends to learn more common watermarked sequences first seems reasonable. If a defender tries to evade the spoofing attack by ignoring the most common watermarked sequences or n-grams, they could reduce the proportion of texts generated by the adversary which are detected as watermarked, but this would also reduce the proportion of texts genuinely generated by original model which are detected as watermarked. This is because the most common watermarked sequences/n-grams contribute more towards the watermark detection score. Rare watermarked sequences are less likely to appear, so it is more likely that a genuinely generated text contains none or few rare sequences, thus resulting in a false negative. So this attempt to evade the spoofing attack will reduce the watermark’s efficacy to detect model-generated texts, which is its primary purpose.
>
> **Spoofing attacks vs defenses**
>
> More generally, we believe that defenses against spoofing attacks vs. spoofing attacks to overcome these defenses is a security cat and mouse game, with no conclusive winner or answer at this time. Some new defense could later be defeated by a new attack, which is then defeated by a new defense, and so on. Ultimately, potential vulnerability to spoofing attacks is one reason among others suggesting that text watermarking should not be used to conclusively attribute provenance or blame to a specific model. Text watermarking is best suited for its original use case of detecting model-generated text, e.g., for preventing students from cheating on essays or preventing LLM-generated misinformation campaigns.

---

> > ### Comment · Reviewer_metd · 2023-11-22
> > **Thank you for your reply.**
> >
> > Thank you for your reply. I will keep my current positive score.

---

### Official Review · Reviewer_2p7U · 2023-10-31

**Soundness:** 3 good
**Presentation:** 3 good
**Contribution:** 3 good
**Rating:** 5
**Confidence:** 3

**Summary:**

This paper raises a concern about forgery attacks on current LLM watermarking methods. To achieve the attack goal, a distillation-based strategy is introduced. Experiments on three existing LLM watermarking methods demonstrate the threat of such an attack.

**Strengths:**

It is interesting to introduce forgery/spoofing attacks into the recent popular area, namely, LLM.

**Weaknesses:**

1. This paper directly extends distillation strategies on decoding-based watermarking and then poses the limitation of weight-based watermarking. However, no solution is provided. It seems like an experimental report, and it would be better to introduce the specific solution.

2.  The motivation of this paper is there is a limitation of decoding-based watermarking, namely, replacing it with a normal decoder. Is the assumption practical? In practice, for an LLM API, how do we conduct such an operation?

3. For wright-based LLM watermarking, more attacks (besides fine-tuning) shall be considered, such as pruning and further distillation.

4. The paper exceeds 9 pages.

**Questions:**

1. Re-clarify the contribution of this paper, and provide strong results or explanations to support the claim. Furthermore, re-clarify the scenario for better understanding.

2. For decoding-based watermarking, if different users are assigned different keys, will the distillation or spoofing attacks fail?

3. What is the additional computational cost? Evaluate or compare it.

4. Besides decoding-base LLM watermarking, there exist some backdoor-based LLM watermarking methods. Are the findings still suitable for them?

---

> ### Author Response · Authors · 2023-11-21
> **Reply to Reviewer 2p7U**
>
> We thank you for the useful and insightful feedback and for taking the time to review our paper. We address questions and concerns below.
>
> **Clarification of contribution**
>
> Please see our general comment clarifying our contribution. https://openreview.net/forum?id=9k0krNzvlV&noteId=UyaqBwxSNh
>
> **Replacing decoding-based watermarking with normal decoder**
>
> For API-only LLMs, decoding-based watermarking is suitable, since the LLM provider controls generation, and users cannot replace the decoding-based watermarking algorithm with a normal decoder. However, for open LLMs, e.g., Llama 2, users run generation themselves, so they can easily choose not to use a decoding-based watermark algorithm, and many inference libraries do not watermark outputs by default or at all. So, decoding-based watermarking is unsuitable for robust watermarking for open LLMs. Our findings into weights-based watermarking address this key technical challenge towards watermarking for open LLMs, though additional challenges remain, e.g., robustness to further fine-tuning.
>
> **Attacks besides fine-tuning**
>
> We demonstrate that simple fine-tuning removes weights-based watermarking capabilities, showing that developing weights-based watermarking that is robust to modifying model weights is an important unsolved challenge. Further attacks would not change our conclusion.
>
> **Paper length**
>
> Per the ICLR author guide https://iclr.cc/Conferences/2024/AuthorGuide, the optional ethics statement and reproducibility statement do not count towards the page limit. The main body of our paper does not exceed 9 pages.
>
> **Assigning different watermarking keys**
>
> We run additional experiments in Appendix E to investigate the behavior of sampling-based watermark distillation if the samples are watermarked using multiple different keys. We also run different trials varying the number of training samples. The table below shows watermark detection p-values.
>
>                           Number of training samples
> | Model | 20k | 40k | 80k | 160k | 320k | 640k |
> | --- | --- | --- | --- | --- | --- | --- |
> | KGW $\delta = 2$ (one key) | 4.1e-02 | 1.4e-02 | 2.4e-03 | 5.2e-04 | 5.1e-05 | 3.6e-06 |
> | KGW $\delta = 2$ (two keys) | 1.7e-01 | 1.4e-01 | 7.8e-02 | 5.7e-02 | 2.9e-02 | 1.4e-02 |
> | Aar $k = 2$ (one key) | 7.4e-02 | 9.4e-03 | 3.5e-04 | 1.0e-06 | 5.1e-10 | 2.9e-14 |
> | Aar $k = 2$ (two keys) | 4.2e-01 | 2.3e-01 | 1.2e-01 | 4.0e-02 | 1.0e-02 | 1.2e-03 |
>
> The results show that using two different keys hinders watermark learning, as indicated by the larger p-values. However, learning is not completely prevented, as the p-values are still much lower than 0.5 and decrease as the number of training samples increases.
>
> Note that if multiple watermarking keys are used, then the watermark detection will lose statistical power. Any suspected text will need to be tested using each of the keys in use, so multiple testing correction must be performed, leading to weaker p-values and a higher false negative rate.
>
> **Additional computational cost**
>
> Weights-based watermarking uses standard generation procedures, so there is no additional computational cost at inference time compared to normal LLMs. Compared to decoding-based watermarking, weights-based watermarking has less computational cost at generation time, since there is no need to run an additional decoding-based watermarking algorithm on top of the LLM. Weights-based watermarking does require model training, the computational cost of which varies depending on model size and dataset size, but this cost is one-time and not incurred at inference time.
>
> **Relationship to backdoor-based model watermarking**
>
> Our findings concern the learnability of decoding-based text watermarking methods. Backdoor-based model watermarking is crucially different in both setting and method. Backdoor-based model watermarking is meant to enable detecting if a suspect model was stolen from a victim model, e.g., through leaking model weights. Detection is done by querying the suspect model multiple times on specially chosen inputs, i.e., the backdoored inputs. If the suspect model produces the chosen backdoored outputs on these backdoor inputs (which are chosen to be different from how a normal, non-backdoored model would behave), then the suspect model is detected as stolen. Note that the backdoored inputs are meant to be stealthy and hard to find, and the backdoored model still behaves normally on normal inputs. On the other hand, decoding-based text watermarking enables detecting if a given text was generated by an LLM using any input prompt. Detection is done without using any LLM, only the tokenizer is needed. Decoding-based text watermarking is designed to affect all model outputs regardless of input, so that ideally all generated texts can be detected as watermarked. Because of these crucial differences in both setting and method, our findings are not directly relevant to backdoor-based model watermarking, and vice versa.

---

> ### Author Response · Authors · 2023-11-22
> **End of discussion period**
>
> Dear Reviewer 2p7U,
>
>
> We appreciate all of the valuable time and effort you have spent reviewing our paper. As today is the last day of the discussion period, we gently request that you review our reply and consider updating your evaluation accordingly. We believe that we have addressed all questions and concerns raised, but please feel free to ask any clarifying questions you might have before the end of the discussion period.
>
>
> Best,
>
> Authors

---

### Author Response · Authors · 2023-11-21
**General comment clarifying our contribution**

Reviewers 2p7U, metd, and GhKC have raised clarifying questions about our exact contribution, particularly with regard to watermarking for open models. We give a clarification of our contribution here, and we have revised our paper accordingly.

The focus of our work is empirically investigating the learnability of decoding-based watermarks for language models, i.e., whether a model can learn to generate watermarked text by itself, without using a specialized watermark decoding algorithm. If a model is successfully able to learn to do so, we call this weights-based watermarking. This question of learnability was previously unexplored, and whether existing decoding-based watermarks are learnable was not known. Our main contribution is our empirical findings, namely that models can learn to generate watermarked text with high detectability, although sample complexity is higher for lower-distortion watermarks and hyperparameter settings.

To motivate and demonstrate the importance of our investigation into watermark learnability, we discuss relevant implications for developing watermarking for open models and for the possibility of spoofing attacks. Weights-based watermarking is necessary for developing robust watermarking for open models (e.g., Llama 2) since users can run inference themselves and choose the decoding procedure, so decoding-based watermarking algorithms are easily disabled, or just not enabled by default. So, our work addresses one of the key technical challenges towards developing robust watermarking for open models. Further important technical challenges remain, such as robustness to further fine-tuning, as users can fine-tune open models on their own data. We believe that solving these remaining roadblocks is best explored in future work.

However, weights-based watermarking also has potential use cases that do not depend on being robust to further fine-tuning. For example, weights-based watermarking could be used for watermarking open models which are unlikely to be fine-tuned further by users, such as RLHF instruction-following chat models. In addition, weights-based watermarking simplifies decoding compared to decoding-based watermarking, as there is no need for an additional specialized decoding algorithm. So, weights-based watermarking can easily be deployed into existing LLM infrastructures and optimized generation algorithms, as it just requires loading different model weights.

In addition, watermark learnability raises the possibility of spoofing attacks, where an adversary mimics the watermark of a victim model. The existence of successful spoofing attacks suggests that watermark detection should not be used to attribute text provenance or blame to a specific model. We demonstrate a simple proof of concept spoofing attack to substantiate their possibility. We believe that comprehensively evaluating spoofing attacks and defenses is a cat and mouse security game which is not necessarily conclusively answerable at this time, and would be best explored in detail in future work, but our work provides a foundation for this future work by showing that spoofing attacks can exist in some settings.

In summary, our main contribution is our empirical investigation and results on the learnability of watermarks. We believe that this contribution is significant and new as watermark learnability was previously unexplored, and it was unknown whether existing decoding-based watermarks are learnable. Our findings address a key technical challenge towards developing watermarking for open models and raise the possibility of spoofing attacks. We hope that our work will inform and inspire future lines of research into remaining technical challenges towards watermarking for open models, comprehensively evaluating spoofing attacks and defenses, etc.

---

### Author Response · Authors · 2023-11-21
**Summary of main changes**

We thank all of the reviewers for their valuable and helpful feedback. We have responded to each reviewer individually and revised our paper accordingly. We summarize the main changes below:
- We run additional experiments using Llama 2 7B as both the teacher and student model for sampling-based watermark distillation in Appendix C. (reviewer hxrQ)
- We run additional experiments investigating the sample efficiency of sampling-based watermark distillation in Appendix D. (reviewer metd)
- We run additional experiments on sampling-based distillation if the training samples are watermarked using two different keys in Appendix E. (reviewers 2p7U, metd)
- We run evaluations on additional datasets (Wikipedia articles and arXiv papers) in Appendix F. (reviewer hxrQ)
- We include the AUROC metric for classifying watermarked vs non-watermarked text in full tables in the Appendix. (reviewer GhKC)
- We include a discussion of model watermarking for protection against model extraction attacks in the related work section (Section 7). (reviewers hxrQ, GhKC)
- We re-ran watermark detection evaluation for sampling-based watermark distillation (Section 5.2) to only detect the first 200 tokens under the Llama 2 tokenizer, which is used for detection, rather than the first 200 tokens under the Pythia tokenizer, to allow for fairer comparison between logits-based and sampling-based distillation (Table 1 vs 2). This caused very slight changes in some of the p-values in Table 2, which do not change any of the trends or conclusions.
- We have revised the paper to make our contribution clearer. (reviewers 2p7U, metd, GhKC)

---

### Meta-Review · Area_Chair_SdzQ · 2023-12-06

**Metareview:**

"On the Learnability of Watermarks for Language Models" provides a comprehensive set of experiments investigating whether recently proposed watermarking strategies for large language models are learnable. The learnability of watermarking strategies has implications both concerning the ease of imprinting a watermark into open-source models and concerning spoofing attacks against watermarks.

Overall the reviewers were positive about this work, although several concerns were brought up concerning the depth of the experimental investigation. The authors adressed these concerns during the response period and I ask them to make sure these additional experiments are included in the camera-ready version of this submission.

As a final remark, I am slightly surprised that the authors did not vary $k$ for the KGW watermark, given that this quantity is central to the learnability of that watermark. Especially in light of related work that is already investigating alternative choices for $k$, such as $k=0$ in Zhao et al., "Provable Robust Watermarking for AI-Generated Text", and $k=4$ in KGW2023b, in the context of other analysis. I firmly believe that the inclusion of these experiments would further strengthen this work and would be of interest to the ICLR community. The choice of  $k=0$ in Zhao et al., for example, is, intuitively, easily learnable.

**Justification For Why Not Higher Score:**

Reviewers considered the topic at hand to be somewhat niche, so I am not proposing this submission for a talk, but I am confident that there is a community at ICLR that would be very interested in discussing the results of this work.

**Justification For Why Not Lower Score:**

Interesting, detailed investigation of a very relevant question, aiding the community's understanding of the pros and cons of different watermarking strategies.

---

### Decision · Program_Chairs · 2024-01-16

Accept (poster)